# Selectivity of Lewy body protein interactions along the aggregation pathway of α-synuclein

André D. G. Leitão [1,3], Paulina Rudolffi-Soto [1], Alexandre Chappard [1,4], Akshay Bhumkar[1,5], Derrick Lau [1], Dominic J. B. Hunter[1,2], Yann Gambin [1✉] & Emma Sierecki [1✉]

The aggregation of alpha-synuclein (α-SYN) follows a cascade of oligomeric, prefibrillar and fibrillar forms, culminating in the formation of Lewy Bodies (LB), the pathological hallmarks of Parkinson's Disease. Although LB contain over 70 proteins, the potential for interactions along the aggregation pathway of α-SYN is unknown. Here we propose a map of interactions of 65 proteins against different species of α-SYN. We measured binding to monomeric α-SYN using AlphaScreen, a sensitive nano-bead luminescence assay for detection of protein interactions. To access oligomeric species, we used the pathological mutants of α-SYN (A30P, G51D and A53T) which form oligomers with distinct properties. Finally, we generated amyloid fibrils from recombinant α-SYN. Binding to oligomers and fibrils was measured by two-color coincidence detection (TCCD) on a single molecule spectroscopy setup. Overall, we demonstrate that LB components are recruited to specific steps in the aggregation of α-SYN, uncovering future targets to modulate aggregation in synucleinopathies.

[1] EMBL Australia Node in Single Molecule Science and School of Medical Sciences, The University of New South Wales, Sydney, NSW, Australia. [2] Institute for Molecular Bioscience, The University of Queensland, Brisbane, QLD, Australia. [3] Present address: Department of Molecular Medicine, The Scripps Research Institute, La Jolla, CA, USA. [4] Present address: School of Chemistry, The University of Edinburgh, Edinburgh, UK. [5] Present address: Woolcock Institute of Medical Research, University of Sydney, Sydney, NSW, Australia. ✉email: y.gambin@unsw.edu.au; e.sierecki@unsw.edu.au

Protein aggregation with formation of intracellular inclusions is a common trait of many neurodegenerative diseases. In Parkinson's disease (PD), these inclusions are named Lewy bodies (LBs), after the German neurologist Friedrich Lewy who first described them in the midbrain[1]. LBs are the pathological hallmark of PD and dementia and can also occur in a series of other neurodegenerative disorders, including Alzheimer's disease and Down's syndrome[2]. The key event in the formation of LBs is the aggregation of their main constituent, alpha-synuclein (α-SYN), an intrinsically disordered protein that is abundantly expressed in neurons[3]. α-SYN aggregates can also be found in multiple system atrophy and neuroaxonal dystrophies[4].

Enhanced oligomerization was first demonstrated by mutations in the α-SYN gene (SNCA) (A30P, E46K, H50Q, G51D and A53T) in vitro and in vivo[5]. These were the first genetic links to PD to be discovered, leading to early-onset PD (patients develop symptoms in their mid-30s). Importantly, recent studies revealed that these disease-associated mutants show very different aggregation kinetics, despite causing similar clinical manifestations[6]. The mutants A30P and G51D mainly form oligomers of a smaller size that incorporate wild-type (WT) α-SYN but do not readily form pre-fibrils; on the other hand, A53T forms larger-size oligomers, which do not recruit WT but resemble pre-fibrils[7].

In 2007, Wakabayashi et al.[8] published a comprehensive review that listed all the proteins identified as constituents of LBs. Until then, >70 proteins had been identified, belonging to 10 different protein classes and involved in a variety of cell functions: structural elements of the LB fibrils, α-SYN-binding proteins, synphilin-1-binding proteins, components of the ubiquitin–proteasome system, proteins implicated in oxidative stress, chaperones, kinases, cytoskeletal proteins, cell cycle proteins, and others.

While many of these proteins are known interactors of α-SYN, the nature of these interactions is still largely unknown. Indeed, as an intrinsically disordered protein, α-SYN's function and ability to recruit binding partners is driven by its conformation and degree of oligomerization. The ability to distinguish properties of molecules depending on their self-assembly status is a crucial factor to understand the driving mechanisms of neurodegeneration, where the formation of higher-order assemblies is the norm. With this in mind, we set out to investigate whether the proteins present in the LBs are trapped or pulled into the aggregates by binding to a particular stage of the aggregation of α-SYN. Most of these proteins are currently assumed as bona fide interactors of α-SYN; however, which components are directly recruited by α-SYN, rather than being recruited by misfolded aggregates or other components, within LBs is for most cases unknown. A thorough description of the interactome of α-SYN along its aggregation cascade was long overdue.

Here we implemented single-molecule methods to study co-aggregation and binding events between α-SYN and other proteins present in LBs. To answer the specific challenge of detecting weak and transient interactions, single-molecule fluorescence methods were applied to track protein–protein interactions in completely undisturbed samples. We performed measurements directly in the translation reaction of the Leishmania tarentolae cell-free protein expression system, bypassing steps of purification or labelling that can modify the oligomerization status of a protein or disrupt interactions.

This work presents strong evidence to the fact that self-assembly of α-SYN dictates its repertoire of binding partners. Proteins in the LBs bind with exquisite specificity to different conformers of α-SYN: we defined interactors of monomeric, oligomeric or fibrillar α-SYN and built a global interactome of α-SYN along the aggregation pathway.

## Results

**Establishing the interactome of α-SYN**. The first step in investigating the interactome of α-SYN was to select a list of candidate partners based on our current understanding of LBs' composition. Here we chose a subset of 65 components that were grouped by functional categories based on the classification established by Wakabayashi et al.[8]. As expected, these include many components of the proteostasis network, such as molecular chaperones, proteins involved in oxidative stress, the autophagy–lysosomal pathway and the ubiquitin–proteasome system. Structural and signalling proteins are also included (Fig. 1a). Open reading frames (ORFs) encoding these proteins were cloned into the appropriate vectors to allow for expression of N-terminal green fluorescent protein (GFP)-tagged proteins using a Leishmania tarentolae-derived cell-free expression system (LTE), thus bypassing the need to purify proteins (Fig. 1b). Protein expression levels were analysed using sodium dodecyl sulfate-polyacrylamide gel electrophoresis (SDS-PAGE; Fig. S1 and Table S1).

To measure protein interactions at the different aggregation stages, we used a combination of assays: AlphaScreen for monomers and single-molecule fluorescence for oligomeric/fibrillar interactions.

**AlphaScreen reveals LB-binding partners of monomeric α-SYN**. Direct interactions between monomeric N-terminal mCherry-tagged α-SYN and the N-GFP LB proteins were assessed using AlphaScreen. AlphaScreen is a proximity assay, where the two interacting proteins bring a "donor" bead and an "acceptor" bead in close proximity. When that happens, the donor beads (coated with the GFP-LB protein) transfer an excited oxygen singlet to the acceptor beads (coated with mCherry-α-SYN) upon excitation at 680 nm. Reaction with thioxene derivatives in the acceptor bead lead to emission of light at 520–620 nm (Fig. 2a). The technique has high sensitivity (up to high micromolar) due to the local increase of the concentration of the proteins at the surface of the beads, allowing the formation of low-affinity complexes[9].

First, mCherry-α-SYN and its GFP protein partners were co-expressed in LTE. Co-expression of the proteins increases the probability to detect an interaction. It also allows to detect the formation of heterodimers, when proteins have a tendency to homodimerize[9]. Pairwise interactions were then tested by AlphaScreen straight from the cell-free extracts, without any purification steps that could perturb weak complexes. AlphaScreen signal is dependent on the concentration of the protein: at optimal protein-to-bead ratios (when the beads are fully coated), a maximum signal is detected as shown by a typical "hook effect" in the quantified luminescence emitted (see Fig. 2b, Methods and Supplementary Methods). Therefore, we performed four serial dilutions of the samples containing the co-expressed proteins to find the optimal proteins-to-beads ratio. Typically, optimal coating of the beads is achieved when we dilute the LTE by 2–3 orders of magnitude. Because of the high range of concentrations that these dilutions encompass, and since the assay was performed in quadruplicate, false positives should be minimal. Furthermore, at these lower concentrations, dissociation rates are high and even aggregation-prone proteins will be largely monomeric. Therefore, the presence of few aggregates should not overwhelm the system and interactions between aggregates will have the same influence as interactions between two monomers. AlphaScreen signals from repeat experiments were averaged (Fig. S2) and the maxima were normalized to background to give a binding index (BI) between 0 (no

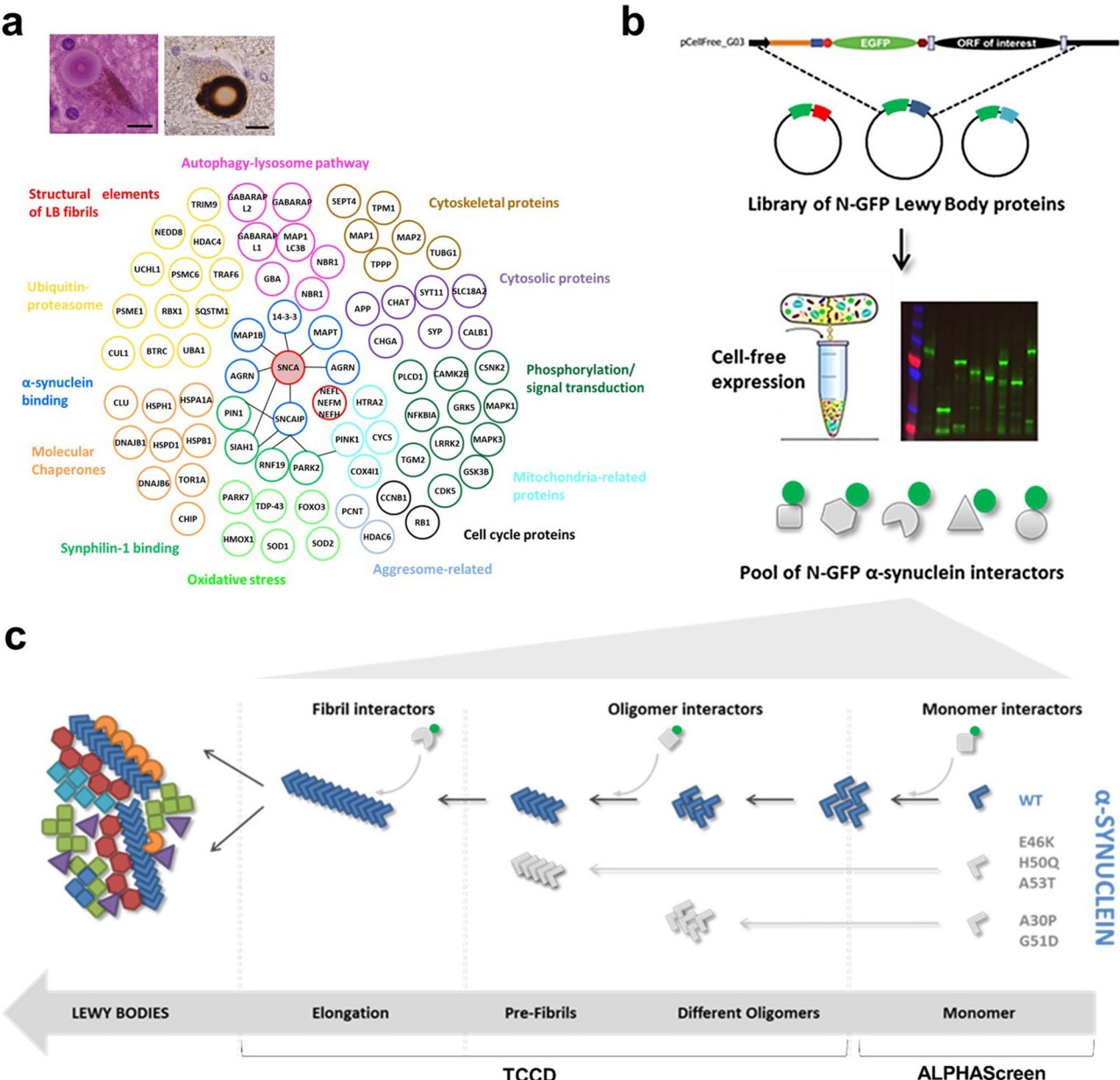

**Fig. 1 Experimental set-up to understand the formation of the Lewy bodies by measuring protein–protein interactions along the aggregation cascade of α-SYN. a** The Lewy bodies contain >80 different proteins, which are involved in several cell processes. Inset adapted from Brown et al.[9]. **b** By using a Gateway cloning system, a library of N-terminal GFP-tagged Lewy body proteins was acquired for expression in the cell-free *Leishmania tarentolae* lysate. **c** On pathway to its inclusion in the Lewy bodies, the aggregation of α-SYN is known for comprising different steps. Upon misfolding of its monomeric form, α-SYN self-assembles into oligomeric, pre-fibrillar and fibrillar forms, respectively. Three different assays were designed to identify the main interactors of α-SYN in four of its forms along this cascade: monomeric (AlphaScreen), oligomeric, pre-fibrillar, and fibrillar (two-colour coincidence detection using a single-molecule spectroscopy set-up).

interaction) and 1 (maximal interaction) (Fig. 2c). Most proteins were found to be only weak binders of α-SYN; however, three proteins emerged as interactors of N-mCherry-tagged α-SYN, when statistically compared to the negative control of monomeric GFP. These interactors include two kinases: G-protein receptor kinase 5 (GRK5) and mitogen-activated protein kinase 1 (MAPK1); and a component of the ubiquitin–proteasome system: proteasome activator complex subunit 1 (PSME1) (Fig. 2c, d). As a control, α-SYN itself was identified as a binder, in agreement with our previous data[7]. We note that the hit rate of the AlphaScreen experiment is much lower than typically observed in our previous protein–protein interaction screens[9–11].

**Single-molecule fluorescence reveals selective recognition of LB proteins by oligomeric α-SYN.** To rapidly access interactions at the oligomeric level, we performed two-colour coincidence detection (TCCD) experiments. In this assay, both free-floating GFP- and Cherry-tagged proteins are detected by fluorescence confocal spectroscopy. Single-molecule fluorescence spectroscopy relies on the detection of fluorescent proteins diffusing in and out of the confocal volume of a microscope (1 fL in volume). In literature, two methods have been used: fluorescence correlation spectroscopy (FCS) and single-molecule spectroscopy. While FCS is extremely powerful in correlating diffusion time with size at very small timescales, it is not optimal when the sample is

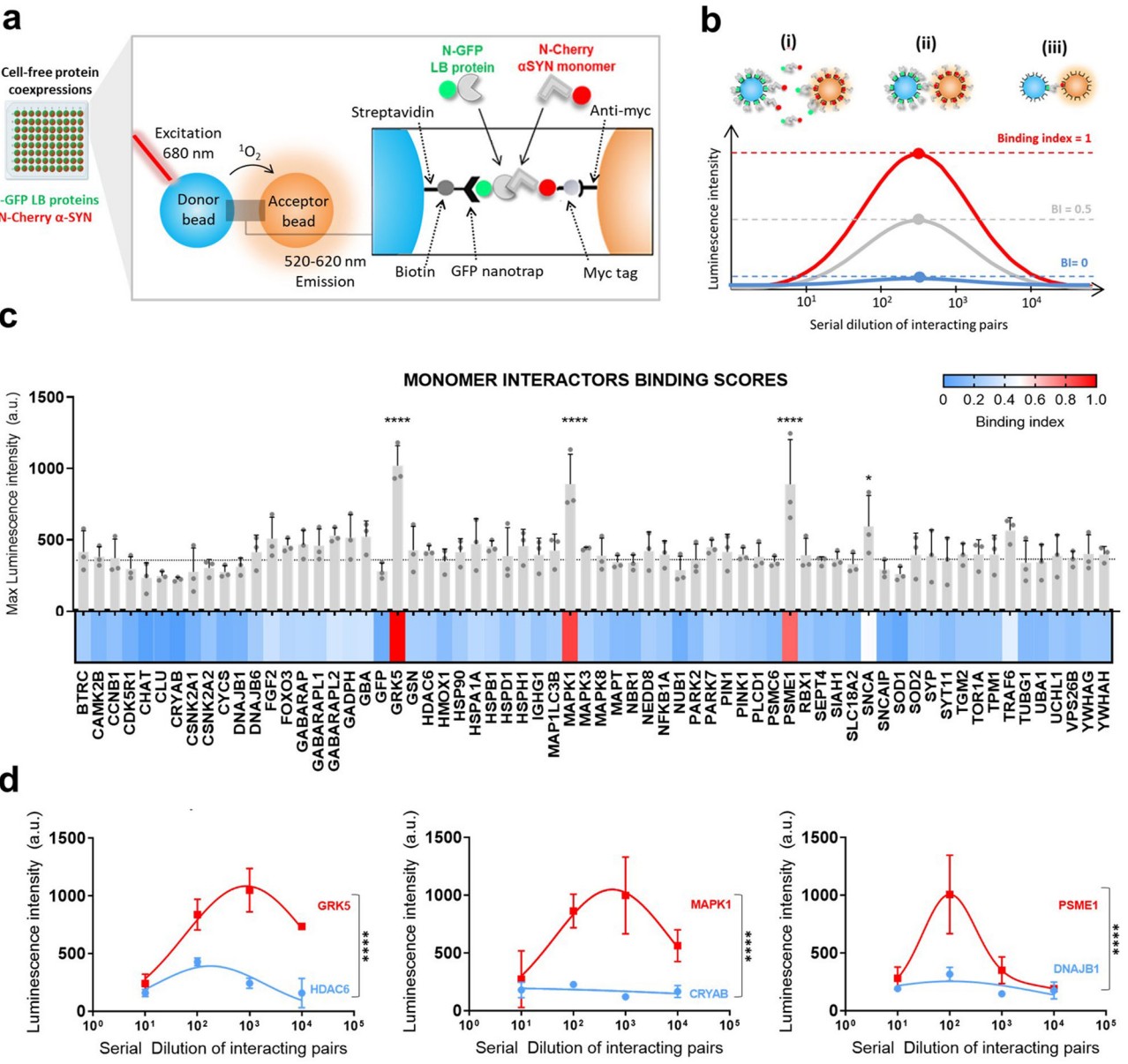

**Fig. 2 Interactions between LB proteins and monomeric WT α-SYN. a** All proteins and a GFP control were co-expressed in LTE with N-mCherry-tagged WT α-SYN. LTE was primed with the DNA constructs of each LB protein and α-SYN in a 96-well plate. The donor bead binds the N-GFP-tagged LB protein while the acceptor bead binds to the N-terminal mCherry-myc tag of α-SYN. Upon interaction, the proteins will bring the beads in close proximity with the transfer of a singlet oxygen, leading to signal being emitted at 520–620 nm. **b** AlphaScreen signal is dependent on the dilution of the protein. An excess of proteins (i) will lead to a low signal by inhibition of bead association through competition with unbound proteins, whereas low concentration leads to limited bead association (iii). A maximum signal is detected when the optimal bead/protein ratio is reached (ii). Maximum luminescence is then converted to a binding index (BI) for each interaction. **c** Maximum values were plotted as a bar plot, with error bars representing the SEM of the triplicate measurements. Dotted line represents mean + SEM of GFP control. Heatmap represents the BI values. **d** AlphaScreen curves along the four dilutions. Comparison of signals to those of the GFP control (Dunnett's multiple comparisons test; ****$p \leq 0.0001$, *$p \leq 0.05$, $n = 3$). Error bars = mean ± SEM.

heterogeneous, as bursts of very large amplitude create long-range temporal correlations. "Pure" single-molecule fluorescence is useful when collecting enough events but are typically performed at pM concentrations, rendering rare events virtually undetectable. Here we used a hybrid approach to overcome some of these limitations by simply analysing the heterogeneity of fluorescence in short time traces, in a background of monomer, performing single particle counting at nanomolar concentrations typically used in FCS experiments.

On the fluorescent time traces acquired in this manner, the diffusion of a large oligomer through the confocal volume results in the appearance of a fluorescent burst well above background.

The presence of two such intense fluorescence peaks in both GFP and Cherry channels identifies as a co-assembly event. We utilized a previously described method for automating the scanning of thresholds for each trace, which proved optimal in eliminating noise and detecting a maximum number of events[12]. This method is described by Clarke and colleagues[12] and was used throughout this study (Fig. S3). The details of the technique are further detailed in Supplementary Methods.

To access the different oligomeric species, we made use of the pathological mutants of α-SYN, which we previously reported to form oligomeric species with distinct properties[7]. We previously observed that A30P and G51D mainly form oligomers of a

smaller size that incorporate WT α-SYN. On the other hand, A53T forms larger-size oligomers, which do not recruit WT. Moreover, the inter-molecular FRET was different between the two groups of mutants, indicating differences in conformations[7]. Analysis of the maximal intensity and residence time of each fluorescent burst shows that the objects formed by the different mutants are relatively homogeneous and confirms that A30P aggregates are less fluorescent (and therefore contain less proteins per aggregates) than the ones formed by G51D and A53T (Fig. S4). Before carrying out binding assays, the mCherry-tagged α-SYN mutants A30P, G51D and A53T were expressed in LTE and observed to behave as the C-GFP versions of the proteins[7,13].

To maximize the chances of detecting coincident events, α-SYN mutants and the candidate binders were co-expressed. To detect rarer species (that could not be detected when relying solely on Brownian motion), the plate holder of the microscope was adapted to move at a constant speed during acquisition (see "Methods"). In this way we detected sufficient events, even in a background excess of monomer. Using this "scanning-plate" set-up, four time traces were acquired for all co-expression. The binding was here quantified by the Association Quotient "$Q$" that measures the ratio of coincident events over the total number of events and takes into account the random chance of co-diffusion, by optimizing threshold selection. Co-expression with mono-meric GFP was used as a negative control, while co-expression with the N-GFP version of the same protein served as positive control (panels i and ii in Fig. 3a, respectively). Here we assume that co-expressing the same synuclein with two different fluorophores provides us with the maximum $Q$. This maximum was similar for the three mutants tested ($Q \sim 0.5$), meaning that, for each threshold, approximately half of the protein assemblies detected contained both fluorophores. The other half was assumed either non-coincident or coincident due to chance—chance of co-diffusion is here designated by $E$ (Fig. 3a, panel iii).

The three mutants of α-SYN presented similar interactomes (Fig. 3b–d). One of the first observations is that the three main hits from the monomeric interactors do not bind significantly to any of the oligomers. The main interactors of the oligomer species showed average $Q$ values between 0.4 and 0.5 (similar to the positive control) and did not bind significantly to the monomeric species, already suggesting selectivity in binding. Interactors of oligomeric species included members of the GABARAP and LC3 (MAP1LC3B) subfamilies, all of which are autophagy-related proteins named ATG8 (traces in Fig. S5). The small heat-shock protein (HSP) αβ-crystallin (CRYAB), a molecular chaperone, showed high coincidence values as well, whereas other chaper-ones did not bind to α-SYN oligomers in our assay (see $Q$ values for HSPB1, HSP90, HSPA1A and HSPD1). All the oligomeric species tested here showed high enrichment in the retromer complex subunit VPS26B (vacuolar protein sorting 26 homologue B). In our system, mutant oligomers were not observed to interact with cytoskeleton elements such as tau or tubulin, but the actin-binding protein gelsolin (GSN) was a statistically significant binder for all oligomeric species.

Most of the binding occurs in sub-stoichiometric ratios, meaning that, in the co-aggregates, α-SYN is the main component. We measured the number of interactors binding to each oligomer by calculating the ratio of the signals in the two channels. In Fig. S6a, we show that, for the protein concentrations used, apparent stoichiometry did not vary significantly. Our data show that only CRYAB formed complexes of higher stoichiometry with α-SYN. This difference is likely related to the propensity of CRYAB to aggregate. Other interactors such as GABARAP and MAP1LC3 are predominantly monomeric and showed low presence in heterogeneous aggregates with α-SYN.

**Co-aggregation versus binding to pre-formed aggregates**. One of the main question is whether the binding occurs as a co-aggregation process during formation of the oligomers or via recognition of a new conformer after the oligomers are fully formed. These two mechanisms can be differentiated by either mixing samples during expression or mixing samples post-expression.

First, we performed experiments in which interactors and α-SYN were mixed after individual expression, thereby testing binding to pre-formed oligomers (Fig. 4a, b). In these conditions, few interactions maintained the same $Q$ values (Fig. 4c). To further validate that a subset of proteins were bona fide oligomer interactors, we generated and purified oligomers from recombi-nant WT α-SYN, according to Kumar et al.[14]. Oligomers were separated by size-exclusion chromatography and characterized by electron microscopy. To test binding to these unlabelled objects (Fig. 4d), we observed changes in the aggregation behaviour of the interactors and quantified the brightness of the GFP-labelled interactor before and after addition of unlabelled oligomers[15]. Binding to oligomers was detected by the apparition of fluorescent bursts in the GFP channel shortly after mixing (Fig. S7). As shown in Fig. 4e, we observed an increase of brightness for GABARAP, GABARAPL1 and GABARAPL2 upon mixing with oligomers. Addition of oligomers had no effect on the controls GFP, CHAT and PARK7, which were not identified as oligomer interactors.

Actual recognition of pre-formed and stable α-SYN oligomers seems limited to proteins from the GABARAP/LC3 family that are involved in the enclosure of protein aggregates by autophagosomes during autophagy. Differences in $Q$ values were statistically significant for all other binders as shown in Fig. 4c; hence, recognition of these proteins occurs during α-SYN oligomerization. This suggests that most of the interactions detected were due to incorporation during aggregate formation (Fig. 4a). Co-aggregators of α-SYN mutants include the small HSP CRYAB, the retromer protein VPS26B, GSN, histone deacetylase 6 (HDAC6), as well as synphilin (or synuclein-interacting protein (SNCAIP)) and α-SYN itself, all of which lose their ability to recognize α-SYN when aggregates are pre-established.

Note that, in physiological conditions, oligomeric species would form in the presence of the various protein interactors, so the co-aggregation process is biologically relevant. The fact that GABARAP/LC3 proteins have an additional mode of recognition make them strong candidates for further studies.

**Pre-formed fibrils (PFFs) of α-SYN show their own inter-actome in the LBs**. We then proceeded to investigate the inter-actions between LB components and amyloid fibrils of α-SYN. To this end, we used bacterially expressed recombinant α-SYN to generate fibrils (Fig. 5a). These were labelled post-aggregation with the fluorescent dye Alexa594 (in a 1 in 10 ratio) and added to cell-free extracts that were expressing the different binding partners tagged with GFP. Again, TCCD was used to determine binding to the fibrils. In this case, due to the presence of larger and more frequent events (as compared to the cell-free expression of mutant synucleins), a single trace of 300 s was acquired for each replicate (Fig. 5b) using a stationary plate set-up.

Overall, we observed a higher apparent binding of LB proteins, including many proteins that did not co-diffuse with mutant α-SYN oligomers (Fig. 5c). Examples are cytochrome C (CYCS) and chaperones from the DNA-J/Hsp40 family (DNAJB1 and DNAJB6), as well as microtubule-associated protein tau (MAPT) (Fig. 5d). Notably, tau was one of the main hits for fibrils and the lowest for oligomeric mutant α-SYN. Further, fractions bound to α-SYN fibrils (Fig. S8) showed considerable differences between

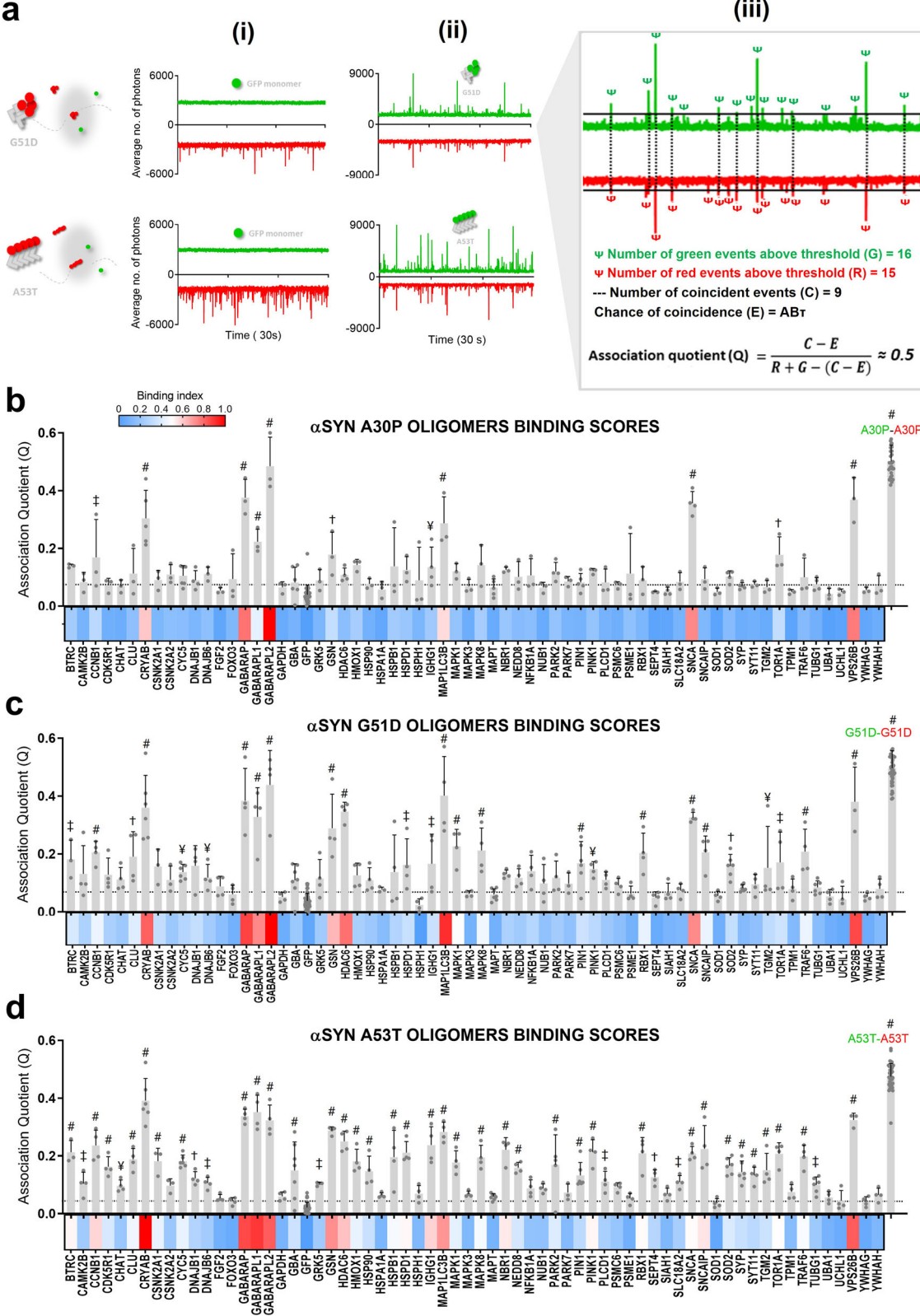

interactors; indeed, when expressed at similar levels, interactors such as members of the GABARAP subfamily showed low values of relative presence of GFP species in the aggregates (between ~0.14 for GABARAPL1 and 0.32 for GABARAP), whereas this value was ~0.5 for other hits, such as CYCS. This difference is also noticeable when comparing the traces for co-expression of these proteins in Fig. 5d.

An absolute comparison of binding affinities between oligomers and fibrils is difficult. The fact that the apparent number of binders seems higher than for the oligomeric species could be due to the higher concentration of α-SYN aggregates in the assay: fibrils are mixed with interactors at 1 micromolar concentration (monomer equivalent concentration). Also, as the fibrils are larger than oligomers, it becomes easier to detect significant

**Fig. 3 Interactors of oligomeric forms of alpha-synuclein. a** The pathological mutants of α-SYN (A30P, G51D and A53T) were tagged with mCherry in their C-termini and co-expressed in LTE with the library of LB proteins. Shaded circles represent the confocal volume (1 fL). Four 30 s traces were acquired for each co-expression and TCCD shows binding to synuclein aggregates. For all traces, GFP signal is represented in positive axis, mCherry signal in negative axis. (i) A negative control of superfolded GFP was used. (ii) The same mutant form of synuclein, C-GFP tagged, served as a positive control. (iii) Association Quotient (Q) was calculated for each trace as the number of coincident events (C) over the total number of events, taking chance coincidence (E) into account. Averages of four measurements were acquired and plotted as bar plots and heatmaps for: A30P (**b**), G51D (**c**) and A53T (**d**). Dotted line represents mean + SEM of GFP control. Dunnett's multiple comparisons test against co-expression with GFP. #$p \leq 0.0001$, †$p \leq 0.001$, ‡$p \leq 0.01$, ¥$p \leq 0.05$, $n = 4$. Error bars show mean ± SEM. Proteins are displayed in alphabetical order.

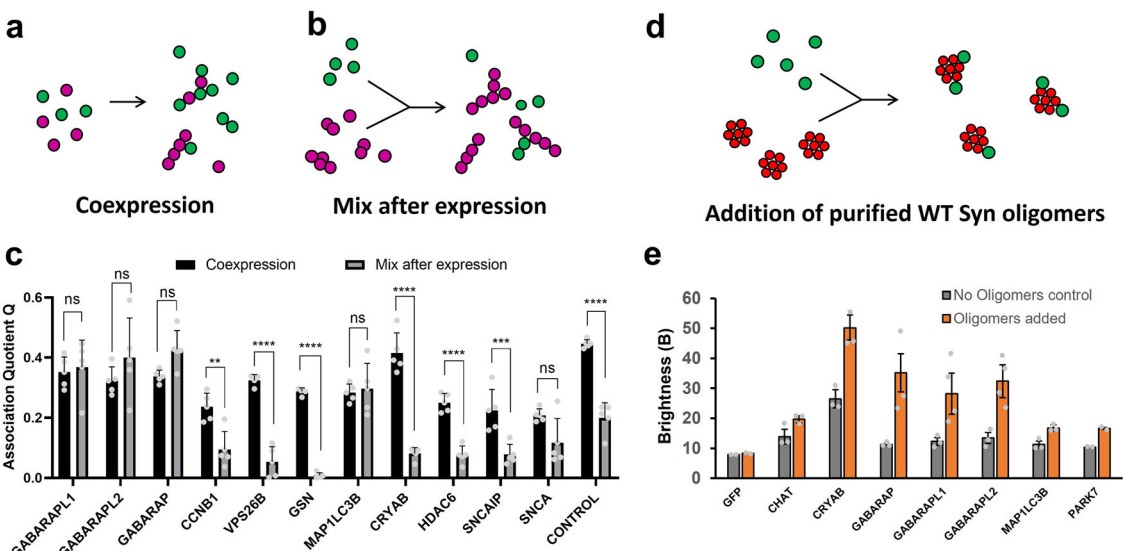

**Fig. 4 Mixing after expression levels reveals binders of pre-formed aggregates in LTE cell-free protein expression system. a, b** Cartoons depicting co-expressing interactors (**a**) versus mixing interactors with pre-formed aggregates of α-SYN (**b**). Red/Green circles represent mCherry-tagged A53T α-SYN and GFP-tagged Lewy body component, respectively. All experiments were performed with cell-free (co)expression of A53T α-SYN. **c** Bar plots of Q values with comparison of means using a two-way ANOVA–Dunnett's multiple comparisons test. ****$p \leq 0.0001$, ***$p \leq 0.001$, **$p \leq 0.01$, *$p \leq 0.05$, $n \geq 4$. Error bars show mean ± SEM. **d** Cartoon depicting mixing interactors with purified oligomers of α-SYN. Red circles represent unlabelled WT α-SYN oligomers and green dots represent GFP-tagged Lewy body components. **e** Bar plots of brightness values of the GFP-tagged interactors in the presence (orange) or absence (grey) of purified oligomers.

binding as measured in the TCCD experiment. However, as the labelling efficiency of α-SYN amyloid fibrils with AlexaFluor594 was 10%, the real number of binders per α-SYN is therefore lower for fibrils than for oligomers. This make sense as the packing of the amyloid core is very dense and each protein binder would physically occupy a space that would cover multiple α-SYN. Retrospectively, this suggests that binding/co-aggregation with oligomeric species was extremely efficient in our system.

**Concentration effects on Association Quotients.** To check that the expression levels of the different interactors did not create a significant bias in our ranking of interactions, we plotted average Q values against the expression levels of the GFP partners, for the oligomer- and fibril-binding assays. Figure S9 shows that there is no discernible trend of higher binding at higher protein expression levels. We noted that some proteins (for example, GSN) displayed relatively high Q values while expressing a lower level than other proteins. This observation led us to simulate the concentration dependence of binding (as estimated by the Association Quotient). The calculation of apparent affinities relies on the assumption of a simple binding mode between interactor and aggregates, as described in Fig. S10 and discussed in Supplementary Note. The simulation of concentration-dependent affinity curves is useful for two reasons. First, it enables us to visualize binders in more detail, revealing some hits that could otherwise be considered non-interactors (e.g. see GSN and

TOR1A for binding to A30P oligomers). Second, this exercise allows us to estimate sensitivities of the oligomer- and fibril-binding assays performed throughout this study. If we consider the threshold of $Q \sim 0.1$ (i.e. the Q value above which there is statistical difference against the GFP alone), we conclude that our binding assays presented a sensitivity of roughly ~1000 nM (fibril interactors) and ~2000 nM (interactors of oligomers) of estimated dissociation constant. This suggests that our oligomer-binding assay was more sensitive, i.e. detected weaker interactors.

The comparison of Q values between oligomers and fibrils reveals important phenomena of selectivity along the aggregation pathway. The raw traces in Fig. S11 show specific interaction of CRYAB for oligomers as compared to fibrils; this is reversed for DNAJB6 where interactions with fibrils is evident but binding to oligomers is undetected. To rule out that interactions detected were due to non-specific "stickiness", we expanded the assay to check their cross-reactivity to non-SYN aggregates.

**Interactome of PTEN-induced kinase 1 (PINK1) in the LBs validates selective recognition of co-aggregates.** To test whether the co-aggregations observed in our system were synuclein specific, we set out to co-express our library of 65 LB proteins to a protein that displayed self-aggregation propensity. One of the best candidates for this cross-reactivity screen is the PINK1. PINK1 is a serine/threonine protein kinase that localizes at both the outer membrane of mitochondria as well as the cytosol. PINK1 scouts

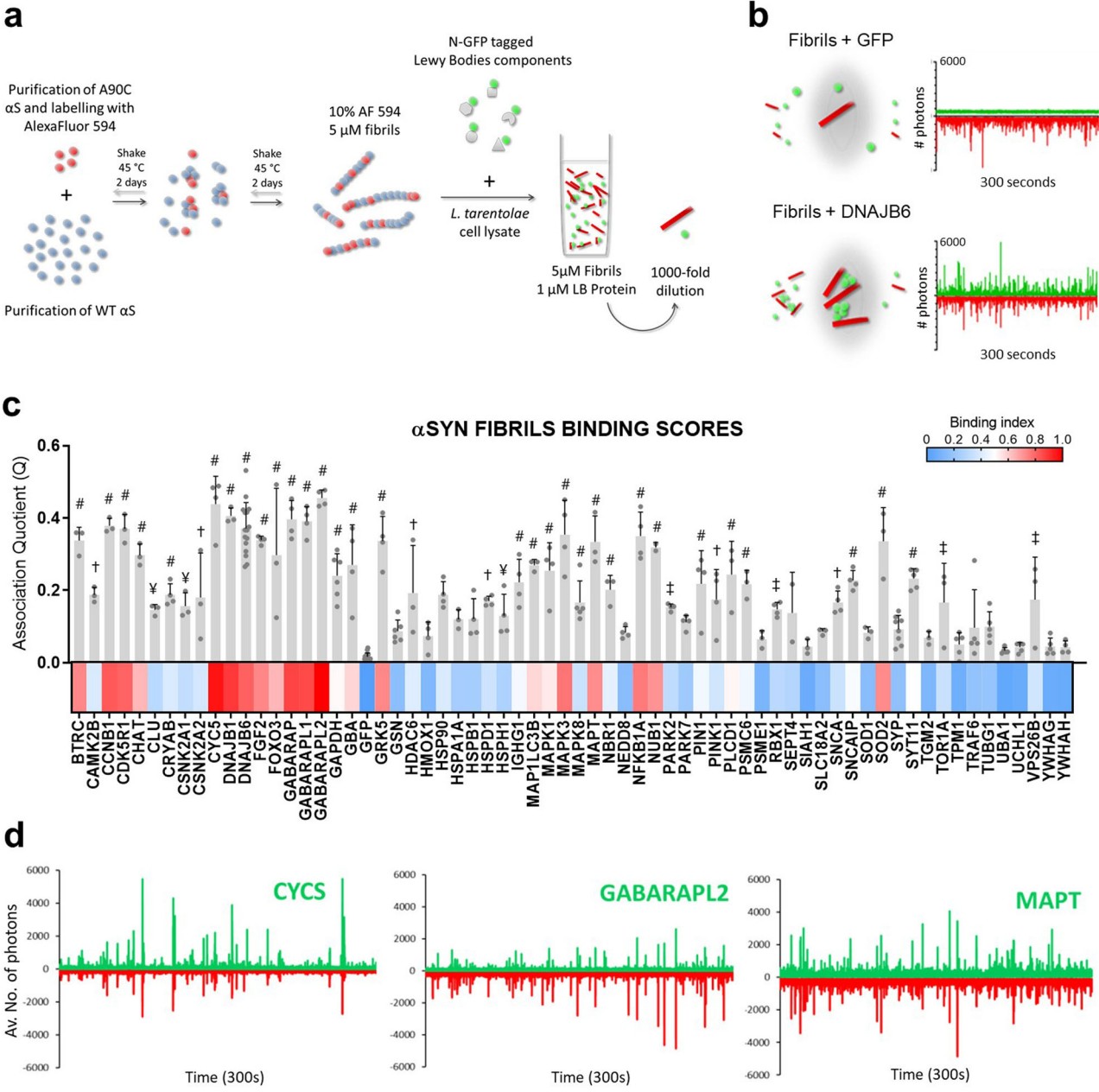

**Fig. 5 Interactors of pre-formed fibrils of alpha-synuclein. a** Schematic representation of the experimental layout for production of pre-formed α-SYN fibrils. Purified cysteine mutated α-SYN was labelled with AlexaFluor dye 594 (red spheres) and mixed with purified WT α-SYN (blue spheres). After shaking at 45 °C, fibrils were primed with the cell-free expression reaction of LTE for all the LB proteins tested. After cell-free expression, the mixture was diluted 1000-fold to reach single particle concentrations for TCCD experiments. **b** Schematic representation of the confocal volumes during measurements of 300 s traces. GFP monomer served as a negative control and a known interactor of αSYN as a positive control (DNAJB6) for acquisition of Association Quotients. **c** Bar plot of $Q$ values for all the quadruplicated 300 s traces and normalized heatmap of binding indexes (BI = 0 for GFP monomer and BI = 1 for the highest $Q$ value). Error bars represent mean ± SEM. Dunnett's multiple comparisons test against traces of Fibrils + GFP. #$p \leq 0.0001$, †$p \leq 0.001$, ‡$p \leq 0.01$, ¥$p \leq 0.05$. **d** Traces of three of the main interactors, as defined by the top percentile in the distribution of $Q$ values. For all traces in this figure, GFP signal is represented in positive axis, mCherry signal in negative axis.

for damaged mitochondria and accumulates at their outer membrane, recruiting Parkin (PARK2), which targets them for degradation through autophagy[16,17]. Mutations in the serine/threonine kinase domain of PINK1 have been found in a number of PD patients, where PINK1 fails to protect against stress-induced mitochondrial dysfunction caused by exacerbated α-SYN aggregation[18,19]. Because PINK1 is genetically implicated in early-onset PD[19,20], accumulates in LBs[21] and forms large aggregates when expressed in our cell-free system, we believed that this would be a good candidate for a selectivity counter-screen. We co-

expressed N-mCherry-tagged PINK1 with the library of 65 N-GFP-tagged LB proteins and measured Association Quotients for all pairwise co-expression, in triplicate. Our data show that the interactome of PINK1 (Fig. 6a) differs significantly from the interactome of a similarly cell-free-expressed synuclein (A53T in Fig. 6b). To understand whether the interactors of α-SYN were selective, we plotted in two dimension the Association Quotients for A53T and PINK1. The scatter plot of Fig. 6b shows that the main interactors of A53T do not bind significantly to PINK1: proteins from GABARAP/LC3 subfamilies, CRYAB and A53T

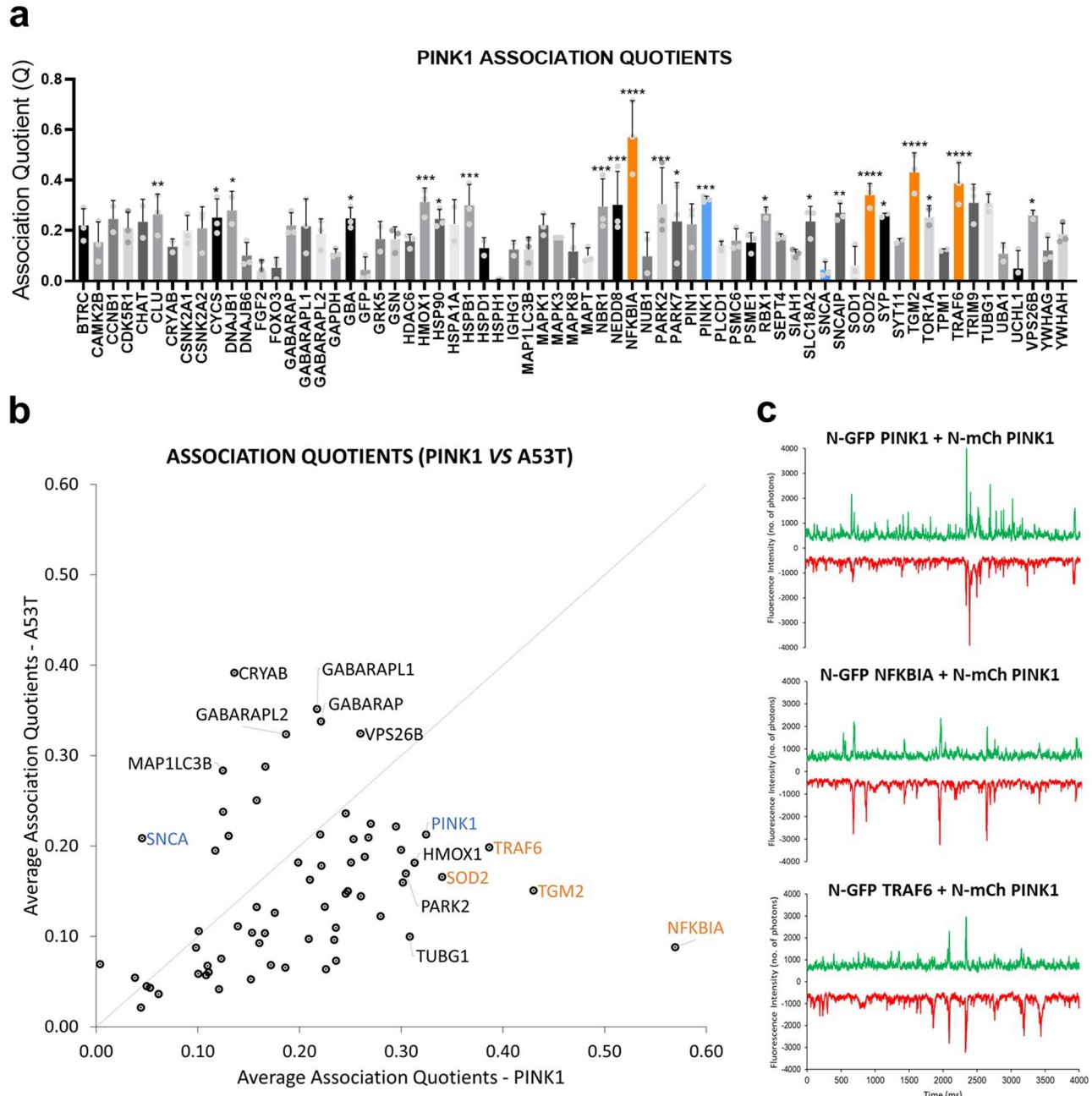

**Fig. 6 The interactome of PINK1 highlights selective recognizers of A53T α-synuclein. a** Bar plot of *Q* values for all triplicated co-expression levels between N-mCherry PINK1 and N-GFP LB proteins (300 s traces). Error bars represent mean ± SEM. Dunnett's multiple comparisons test against traces of co-expression with monomeric GFP. \*\*\*\**p* ≤ 0.0001, \*\*\**p* ≤ 0.001, \*\**p* ≤ 0.01, \**p* ≤ 0.05, *n* = 3. **b** Plots of average Association Quotients for co-expression with A53T (*Y* axis) and PINK1 (*X* axis). Controls for each co-expression are represented in blue (i.e. SNCA-A53T and PINK1-PINK1). Diagonal line represents interactors that were not selective for either A53T or PINK1. Highest *Q* values for interactors of PINK1 represented in orange. **c** Four-second representation of traces for main interactors of PINK1. For all traces, GFP signal is represented in positive axis, mCherry signal in negative axis.

itself have high *Q* values for A53T but low *Q* values for PINK1. The scatter plot also reveals specific interactors of PINK1, such as PARK2[22], TRAF6[23] and TGM2[24] (Fig. 6c), all of which are known interactors of PINK1, suggesting that our co-aggregation experiments can indeed reveal selective binding.

**Global interactome of α-SYN in the LBs.** The combination of the different data sets defines a global interactome of α-SYN in all its forms en route to the formation of LB. The initial pool of interactors was grouped by biological process, according to gene ontology. Information for previously documented interactors was acquired using BIOGRID[25]. Our resulting heatmap of interactions reflects BIs, assuming a BI = 1 for the most significant interactor. Our single-molecule TCCD data reveal a dramatic change in the interactomes, showing that both the oligomeric and the fibrillar state profoundly modify the ability of α-SYN to interact with its environment (Fig. 7). α-SYN fibrils recruit multiple proteins; this fits with the fact that α-SYN fibrils are the main component of the LBs and the ones responsible for inclusion formation[26]. Several interactors, including co-chaperones from the DNA-J/Hsp40 family DNAJB1 and DNAJB6, mitochondrial proteins CYCS and superoxide dismutase 2, MAPT,

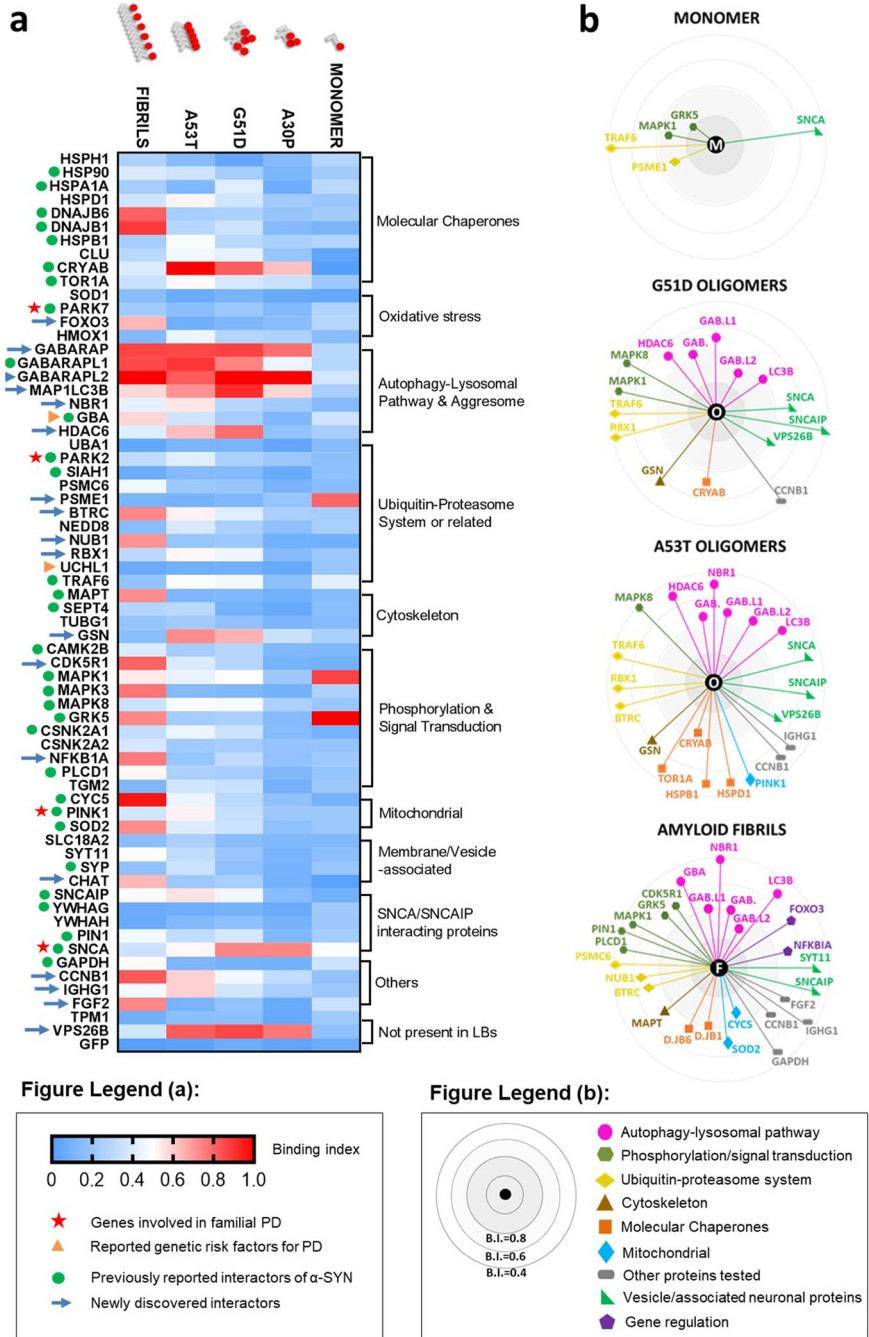

**Fig. 7 The interactome of α-synuclein in the Lewy bodies. a** Heatmap of interactions reflects binding indexes, distributed from the highest (BI = 1) to the lowest (BI = 0), based on Association Quotients (*Q*) from TCCD data and maximum luminescence using AlphaScreen proximity assay. Each colour depicts the average of triplicate measurements of one pairwise interaction with a LB component. Proteins are organized broadly, based on biological process. Previously identified PPIs (as described in BIOGRID) and genetically relevant proteins are highlighted. **b** Binding indexes allowed for representation of individual interactomes for each species of α-SYN tested (M—monomer, O—G51D/A53T cell-free aggregated species and F—amyloid fibrils). Proteins are grouped by biological process and distance to the centre represents magnitude of binding index.

kinases such as NF-kappa-β inhibitor alpha, MAPK3/ERK2 and CDK5R1 (Fig. S12), are specific to fibrillar species. On the other hand, few binders were specific to oligomeric α-SYN: the molecular chaperone CRYAB, GSN, the retromer complex component B (VPS26B), and α-SYN itself (SNCA), which co-diffused more frequently with early aggregates (Fig. S12). Finally, selective recognition of larger oligomeric species was detected for HDAC6, E3 ubiquitin–protein ligase RBX1 and synphilin (SNCAIP) (Fig. S12), suggesting a preferential binding to different conformations of α-SYN.

## Discussion

In this work, we demonstrated a biophysical approach to investigate how self-assembly of α-SYN modulates its ability to bind to different partners. Notwithstanding their invaluable contribution to the identification of protein networks, proteomics studies often neglect the fact that for some proteins the formation of supramolecular assemblies is the norm and can profoundly affect partner selection. α-SYN is a good case study, as it is an intrinsically disordered protein with multiple binding partners that oligomerizes and self-assembles into higher-order structures.

We believe that the main contribution of this study is an unbiased deconvolution of monomeric, oligomeric and fibrillar interactors of α-SYN. Our study, as a purely in vitro approach, does not tell a story of the interactome of synuclein inside the cell, nor does it intend to. It nonetheless suggests that recruitment of protein partners is hierarchical, occurring at specific steps along the aggregation of α-SYN from monomer to amyloid fibrils. This hypothesis, though still controversial, had been proposed before by Betzer and colleagues[27], whose work described a top–down mass spectrometry approach using monomeric and oligomeric α-SYN as baits in lysed fractions of porcine brains. Despite large differences in data sets and approaches, we validated interactions between α-SYN oligomers and *CRYAB* as well as between A53T oligomers and 60-kDa HSP *HSPD1*. Our work expands this data set by showing that pre-formed amyloid fibrils of α-SYN also act as recruiters of proteins (passive or active as this recruitment may be). As summarized in Fig. S13, our results overlap with published interactors and reveal new binders. More importantly, our results suggest that α-SYN oligomers and fibrils are the main contributors to the complexity of protein interactions in LBs, whereas monomeric α-SYN is not as promiscuous an intrinsically disordered protein as previously thought[28]. The study by Betzer and colleagues[27] corroborates this idea, as only ten proteins were pulled-down with the α-SYN monomers, in contrast with 76 interactors of oligomers.

Our global interactome of α-SYN uncovers striking phenomena of selectivity along the self-assembly pathway, which could have a profound impact in the progression of disease. Interactors at specific stages could theoretically enhance (i.e. seed) or rescue aggregation (i.e. attempt to inhibit it). Other interactors could become trapped in inclusions while attempting to inhibit the aggregation of α-SYN. This could explain why molecular chaperones, proteasomal components and autophagy proteins are ubiquitous in brain inclusions, as discussed below.

An example of new insights from our study comes from the interaction between tau (*MAPT*) and α-SYN. Tau and α-SYN are known to synergistically promote fibrillization of each other both in vitro[29] and in cellulo[30,31]. Amyloidogenic proteins such as tau and α-SYN share many conformational similarities (notably β-sheet structures), which could explain their cross-seeding. We report here that interactions between tau and α-SYN occur predominantly when fibrils of α-SYN are already established. Tau monomers can potentially bind along the surface of α-SYN fibrils, which could then act as a nucleator of tau aggregation, in agreement with other authors[29].

The interactome of α-SYN also depends on the cellular context[32]. A recent study using APEX labelling of α-SYN in neurons coupled with mass spectrometry has identified 225 proteins, including synaptic proteins, proteins involved in endocytic vesicle trafficking, the retromer complex, phosphatases and mRNA-binding proteins[33]. A follow-up study showed that endocytic trafficking proteins overlapped with genetic modifiers of α-SYN toxicity, suggesting that perturbation of this pathway is directly related to the spatial localization of α-SYN within the cell[34]. While here we did not test proteins involved in endocytosis specifically, the high interaction scores between α-SYN aggregates (oligomers and fibrils) with proteins from the autophagy–lysosomal pathway suggests that aggregation might act as a disrupter of normal vesicle formation, coalescence, or trafficking. Interestingly, the same study found high enrichment of both MAP1LC3A (homologue of MAP1LC3B) and MAPT[33], which we show here to co-diffuse with pre-formed synuclein oligomers and fibrils.

As mentioned, the primary goal of this study was to show that self-assembly affects protein partner recognition. This in vitro study has limitations (discussed below) as it does not recapitulate the complexity of what is happening in a cellular context from the role of post-translational modifications (PTMs) to the importance of cellular localization and micro-environment. Nevertheless, we believe that it also sheds a light on previous observations in relevant cellular context, due to its inherent simplicity. Examples of such insights are presented below. The importance of these kind of studies has been recently demonstrated by Melki and colleagues[35] who engineered polypeptides to coat the surface of α-SYN fibrils in such a way that their ability to bind to the plasma membrane components and/or their take-up by cell are diminished.

To study the interactome of α-SYN monomers, we first used a proximity assay (AlphaScreen). Because of the nature of the assay, AlphaScreen mainly detects interactions between monomers or small oligomers. Perhaps surprisingly, monomeric α-SYN did not bind to the majority of LB proteins tested, including many previously identified interactors of α-SYN. The hit rate of the AlphaScreen experiment is much lower than typically observed in our previous protein–protein interaction screens[9,11,36]. However, one can also rationalize that the LBs mainly contain interactors of the "pathological" form of α-SYN, while monomers may represent the physiological form of the protein, which could explain the low hit rate observed. As stated before, binding was detected for 2 kinases (GRK5 and MAPK1) and a subunit of the proteasome complex (PSME1) (Fig. 2).

The identification of GRK5 as an interactor of α-SYN is not unexpected based on various observations in cells. Previous studies have shown that membrane-bound-GRK5 not only colocalizes with α-SYN in LBs[37] but also phosphorylates Ser-129 of α-SYN at the plasma membrane. The membrane localization undoubtedly plays a critical role in the interaction between GRK5 and α-SYN but our in vitro data indicate that there is an affinity between the two proteins that contribute to their interaction, beyond co-localization. Phosphorylation of α-SYN then induces its translocation to the perikaryal area of neuronal inclusions[38], consistent with the maturation from pale bodies to LBs. Phosphorylation of α-SYN at Ser-129 has long been identified as a major modification in LBs[39]. More than 90% α-SYN has been found to be pS129 in the brains of patients with synucleinopathies, as opposed to ~4% in healthy brains[39–42]. Importantly, GRK5-catalysed S-129 phosphorylation also promotes the formation of soluble oligomers and aggregates of α-SYN[37]. Previous studies have shown that α-SYN phosphorylation is mainly occurring at the fibrillary level, after inclusions are formed[43,44]. However, our data suggest that GRK5 phosphorylation could also be an early event, occurring at the monomer level, possibly affecting downstream aggregation and/or protein–protein interactions.

Other kinases such as MAPK have not been shown to phosphorylate α-SYN[45], yet they are constituents of LBs[46–49] and glial cytoplasmic inclusions[50]. Studies have reported that α-SYN binds MAPK proteins, namely MAPK1 (ERK1)[51], in agreement with our AlphaScreen findings. It is hypothesized that such interaction reduces the available pool of MAPKS to be phosphorylated by MAPKKs and this is supported by downregulation of MAPK signalling pathways upon overexpression of α-SYN[51]. Activation of MAPK signalling is required for catecholamine[52] and dopamine release[53], a crucial step in synaptic transmission.

One of the most striking phenomena of selectivity observed in our screen was the recognition of α-SYN aggregates by molecular chaperones belonging to different families. Here we analysed interactions with several HSPs, representative of different groups: Hsp100, Hsp90, Hsp70, Hsp60, Hsp40, and small HSPs. Such a screen provides valuable information, particularly because the different HSPs exert their function in a concerted action of chaperone networks[54]. In our experimental set-up, only three out of the ten HSPs tested were clear hits: the small HSP CRYAB,

which showed high *Q* values for all mutant-generated α-SYN oligomers (and low values for fibrils); and two Hsp40/DNAJB family members, DNAJB1 and DNAJB6, which were among the main interactors of α-SYN fibrils.

Oligomers have been postulated to be the most toxic species by several studies in the field and several groups have focussed on understanding the reason underlying this higher toxicity exerted by oligomers[55,56]. Here we have observed some chaperones recognizing early misfolded states (e.g. co-aggregation of CRYAB with α-SYN), while others were the main hits of fibrils (HSP40s), thus the activity of the two is possibly related. It is noteworthy that none of the other chaperones tested co-diffused with α-SYN forms in our assays. In the case of Hsp70s for example, it is well known that they are not recognizers of protein aggregates per se, necessitating an elaborate process of binding and releasing substrates from Hsp40s (co-chaperones)[54,57].

Our TCCD data for co-expression of LB proteins with α-SYN oligomers (Fig. 3) and PPFs (Fig. 5) hint at the importance of autophagy proteins ATG8s (GABARAP and LC3 subfamilies) in the recognition of α-SYN aggregates. Crucially, we have also demonstrated that the interaction with ATG8 proteins is due to binding to pre-formed α-SYN aggregates rather than to the incorporation of ATG8s during aggregate formation (Fig. 4). In sum, ATG8 proteins seem to bind to pre-formed α-SYN aggregates, preferentially in their oligomeric or pre-fibrillar form. One possibility is that such interactions promote packing or clustering of α-SYN conformers and progression towards the formation of inclusions, which could fit with several published reports on the clustering of α-SYN with impairment of macroautophagy[58–60] (Fig. S14).

Interplay between α-SYN and the autophagy pathway has been demonstrated before, in relevant cellular contexts. Tanik and colleagues[58] have demonstrated that α-SYN aggregates interfere with the ability of autophagosomes to mature and fuse to lysosomes. Our data fit with these findings and we propose that binding (possibly clustering) of α-SYN aggregates around GABARAP/LC3 could impair their ability to recruit autophagy adaptors for autophagosome growth, transport along microtubules and lysosomal fusion. The fact that GABARAPs and MAP1LC3B do not interact with monomers and do not co-aggregate with α-SYN (instead binding to pre-formed aggregates —Fig. 4) could corroborate this theory.

Mutant forms of α-SYN associated with early-onset PD show much higher propensity to aggregate and have been shown to destabilize autophagy. Indeed, A30P, G51D and A53T oligomers co-diffused with ATG8s as frequently as mature fibrils (Fig. 7), suggesting correlation between these mutant forms and advanced stages of disease. Other authors have shown that α-SYN aggregates also impair autophagy through inhibition of Rab proteins with subsequent mislocalization of ATG9 and defective formation of early vesicles[59,61]. Importantly, restoration of normal autophagic flux reduced ALP (autophagy–lysosomal pathway) impairment and could be an effective strategy in delaying neurodegeneration[62,63]. Understanding the mechanisms of binding between α-SYN aggregates and GABARAPs/LC3 could offer insights on how to restore normal autophagy fluxes.

Here we provide evidence of direct interactions between members of the GABARAP/LC3 subfamily of autophagosomal proteins and oligomeric/fibrillar α-SYN. Impairment of the formation of autophagosomes by α-SYN aggregates has been shown by several studies, resulting in traffic jams of proteins, organelles and lipid membranes[64–66]. Previous proteomics studies have provided contradictory results as to whether LC3 proteins actually interact with α-SYN[67] and our data suggest that such interactions can only occur when aggregation is under way. Stykel et al.[68] showed direct interaction and sequestration of

MAP1LC3B into microaggregates by A53T and E46K mutants of α-SYN, in agreement with what we observed. Autophagy receptors bind to members of the GABARAP/LC3 family through an LC3-interacting region (LIR) that recognizes an LIR-docking site in the ATG8 proteins[69]. Interestingly, here we report association between α-SYN aggregates and the same ATG8 proteins without involvement of LIR-containing receptors. Although α-SYN does not show the canonical LIR sequences, the requirements for interactions with ATG8 proteins remain controversial. In Fig. S15, we have scanned the list of LB components for the presence of LIR motifs and discussed their potential significance in Supplementary Note. Structural studies using LIR peptides showed that LIRs usually bind as an extended β-sheet to the docking site[69], and β-sheet formation is a well-described critical step in the aggregation of α-SYN[70–72]. Kalvari and colleagues[73] have also suggested that binding to ATG proteins is a conformational switching phenomenon (disorder to order transition), which is characteristic of amyloidogenic proteins and could explain our findings.

In our study, we used the pathological mutants and the PPFs of α-SYN as the representative species of intermediate (oligomeric) and endpoint species of the aggregation cascade, respectively. With respect to the oligomers originated by A30P, G51D and A53T αSYN upon cell-free expression, it is still largely unclear whether these oligomeric structures represent an on- or off-pathway stage in the aggregation process. Therefore, although the study of pathological mutants provided us with an expedite way to acquire the interactome of different aggregated conformers of α-SYN, the feasibility of associating our results with the aggregation pathway of the WT protein is, in large part, speculative. However, to study the interactome of the pathological mutants also brings about possibilities to access pathological folds of synuclein and to understand which protein partners in the LBs could be their targets.

Regarding the in vitro-generated PPFs of α-SYN, despite allowing us to obtain an interactome of α-SYN fibrils with LB proteins, the full biological implications of this interactome are unclear. For example, recent studies describe important differences between fibrils generated in vitro from the ones collected from patient brain samples[74]. Furthermore, various polymorphs of α-SYN aggregates occur in cellulo and in vivo, with different conformations, membrane binding and seeding abilities[75], and sometimes even leading to different pathologies where LBs co-occur[76]. For example, a recent study used liquid chromatography and tandem mass spectrometry (LC-MS/MS) to find at least 20 different forms of α-SYN within patient-derived LBs[77]. This included acetylated, phosphorylated and truncated α-SYN, although their prevalence in disease is still to be determined.

Another limiting aspect of our study is the fact that it is inherently a simplification of protein–protein interaction networks in the highly heterogeneous environment that is the LB. For pragmatic reasons, we tested only pairwise interactions between Cherry- or AF594-tagged α-SYN conformers and the library of N-GFP tagged proteins. This approach does not come without its limitations, as many protein–protein interactions are mediated by cofactors (for example, molecular chaperone Hsp70 that interacts with α-SYN only when it is released as a substrate from Hsp40s) or necessitate the presence of PTMs such as ubiquitination and phosphorylation[67,78].

Finally, the use of GFP and mCherry fluorescent tags attached to one end of the protein is another variable of relative uncertainty, concerning their effect on the interactions that were screened. The relative large size of the tags could theoretically limit the availability of the binding site of a given protein and introduce false negatives into our interactome of α-SYN. Importantly, the presence of a tag should not affect selectivity between the different species of α-SYN. Furthermore, previous

studies have demonstrated that tagging synucleins with GFP does not alter their aggregation behaviour[79], despite the small size of the protein. It is possible, however, that, among all the LB proteins tested, some would see their binding abrogated.

Despite the simplicity of our approach, we believe that our findings provide important clues into the mechanism of inclusion formation, one pairwise interaction at a time.

In conclusion, this study presents an extensive data set to show that, despite its small size and absence of structure, α-SYN binds specifically to different partners along the self-assembly pathway. The single-molecule methods used here enable to observe the formation of co-aggregates when multiple proteins are co-expressed or quantify binding to pre-formed oligomers and mature fibrils. To the best of our knowledge, this is the first study to incorporate the self-assembly of α-SYN into a protein–protein interaction screen at a large scale. Overall, this work provides new insights into the protein–protein interaction landscape along the aggregation cascades and lays the groundwork for future studies on how to modulate α-SYN aggregation at different steps.

## Methods

**Preparation of LTE**. *L. tarentolae* cell-free lysate was produced as described by Hunter et al.[80]. Briefly, *L. tarentolae* Parrot strain was obtained as a LEXSY host P10 from Jena Bioscience GmbH, Jena, Germany and cultured in TBGG medium containing 0.2% v/v Penicillin/Streptomycin (Life Technologies, Carlsbad, CA, USA) and 0.05% w/v Hemin (MP Biomedicals, Seven Hills, NSW, Australia). Cells were harvested by centrifugation at $2500 \times g$, washed twice by resuspension in 45 mM HEPES buffer, pH 7.6, containing 250 mM sucrose, 100 mM potassium acetate and 3 mM magnesium acetate and resuspended to 0.25 g cells/g suspension. Cells were placed in a cell disruption vessel (Parr Instruments, Moline, IL, USA) and incubated under 7000 KPa nitrogen for 45 min and then lysed by rapid release of pressure. The lysate was clarified by sequential centrifugation at $10,000 \times g$ and $30,000 \times g$ and anti-splice leader DNA leader oligonucleotide was added to 10 µM. The lysate was then desalted into 45 mM HEPES, pH 7.6, containing 100 mM potassium acetate and 3 mM magnesium acetate, supplemented with a coupled translation/transcription feeding solution[81] and snap-frozen.

**Gateway cloning to obtain plasmids for cell-free protein expression**. We have obtained the ORFs encoding the desired proteins from IDT. α-SYN point mutants were obtained as gblocks from IDT. A list of LB components was generated based on a comprehensive review from Wakabayashi and colleagues[8] (see Table S1 for all LB proteins used in this study). These genes were sourced from the Human ORFeome collection version 1.1 and 5.1 or the Human Orfeome collaboration OCAA collection (Open Biosystems).

In order to obtain cell-free expression vectors of fluorescently tagged proteins, a gateway cloning method was used, as described elsewhere[82]. Following this protocol, entry clones were generated with PCR primers (Forward primer: 5'GGGGACAAGTTTGTACAAAAAAGCAGGCTT (nnn)$_{18-25}$ 3', Reverse primer: 5'GGGGACCACTTTGTACAAGAAAGCTGGGTT (nnnn)$_{18-25}$ 3')—primers to attB1 and attB2 sites, respectively (sites flanking the inserts). ORFs were then cloned into Gateway destination vectors designed for cell-free expression[83]: N-terminal GFP tagged (pCellFree_G03), N-terminal mCherry tagged (pCellFree_G05), C-terminal GFP tagged (pCellFree_G04), or C-terminal mCherry-cMyc tagged (pCellFree_G08). The successful clones were selected on LB-Ampicillin (100 µg/mL) agar plates, transformed on *Escherichia coli* DH5α competent cells, grown on LB-Amp and the plasmid DNA was extracted using the Presto™ DNA miniprep kits and used for cell-free expression with LTE. Following expression, SDS-PAGE gels confirmed the band size (Fig. S1) and the library of clones was acquired. Finally, all DNA was sequence-verified using Sanger sequencing methods at the Ramaciotti UNSW Center for Cancer and Genomics.

**AlphaScreen assay**. AlphaScreen is a nano-bead-based proximity assay that allows for rapid and sensitive detection of even weak protein-protein interactions on a cell-free-based protein expression system. The method is explained in detail in Supplementary Information.

AlphaScreen cMyc detection and Proxiplate-384 Plus 384-well plates were purchased from Perkin Elmer (MA, USA). LB proteins bearing a N-terminal GFP tag and α-SYN labelled with N-terminal mCherry-Myc were co-expressed in the cell-free system by adding mixed DNA vectors in 10 µL of the *L. tarentolae*-based cell-free system (to a final DNA concentration of 30 nM for the GFP-vector and 60 nM for the Cherry-vector). The mixture was incubated for 3.5 h at 27 °C. Four serial dilutions of the proteins of 1/10 were made in buffer A (25 mM HEPES, 50 mM NaCl). The AlphaScreen Assay was performed in 384-well plates. Per well, 0.4 µg of the Anti-Myc coated Acceptor Beads (PerkinElmer, MA, USA) was added in 12.5 µL reaction buffer B (25 mM HEPES, 50 mM NaCl, 0.001% NP40, 0.001%

casein). In all, 2 µL of the diluted proteins and 2 µL of biotin-labelled GFP-nanotrap (diluted in reaction buffer A to a concentration of 45 nM) were added to the acceptor beads, followed by incubation for 45 min at room temperature. Then 0.4 µg of Streptavidin-coated donor beads diluted in 2 µL buffer A were added, followed by an incubation in the dark for 45 min at room temperature. AlphaScreen signal was recorded on a PE Envision Multilabel Platereader, using the manufacturer's recommended settings (excitation: 680/30 nm for 0.18 s, emission: 570/100 nm after 37 ms).

Overall, three experiments were done for each pairwise interaction. AlphaScreen signals were averaged to obtain Luminescence curves (Fig. S2) and the maximum values of those signals were normalized to background to give the BI. Heatmaps of interactions were constructed based on BIs (Fig. 2b, c).

**Production of PFFs of α-SYN**

*Expression and purification of α-SYN for fibril formation.* α-SYN WT and A90C (vector pT7-7) were transformed in BL21 (DE3) cells and grown at 37 °C in 1 L batches of TB medium with 100 µg/mL Ampicillin. Cells were induced with IPTG for 4 h and harvested by centrifugation at $9000 \times g$, 4 °C for 20 min. The supernatant was discarded, and the pellet was resuspended in 20 mL lysis buffer (100 mM Tris-HCl, 10 mM EDTA, 1× protease inhibitor, pH 8.0) for each litre grown. The cells were lysed by sonication and the lysate was then boiled for 20 min to denature host protein before centrifugation at $22,000 \times g$, 4 °C for 20 min. The supernatant was collected, and 10 mg/mL streptomycin sulfate was added to remove nucleic acids, followed by another centrifugation at $22,000 \times g$, 4 °C for 20 min. The supernatant was again collected, and ammonium sulfate was added at a concentration of 0.4 g/mL to precipitate protein. The mixture was stirred for 30 min at 4 °C before centrifugation at $22,000 \times g$. The resulting pellet was resuspended in a minimal volume of 25 mM Tris-HCl pH 7.7 and dialysed overnight against 20 mM Tris pH 7.7. The protein was purified by anion exchange using an HP/Q Sepharose column (GE Healthcare), which was equilibrated with 2 column volumes (120 mL) of wash buffer (20 mM Tris pH 7.7). The sample was injected onto the column using a 50 mL Superloop (GE Healthcare) and eluted over a NaCl gradient from 0 to 2 M collected in 2 mL fractions. SDS-PAGE allowed suitable fractions to be pooled and concentrated using a 2 kDa filter. This was followed by a step size-exclusion chromatography using Superdex G75 column (GE Healthcare), which was equilibrated with 2 column volumes (300 mL) of buffer (20 mM NaPO4 pH 7.4). The sample was injected into the column using a 10 mL Superloop and eluted over 1 column volume collected in 2 mL fractions. Suitable fractions were collected, checked on SDS-PAGE and concentrated with a 3 kDa filter. Protein concentration was estimated from the absorbance at 275 nm using an extinction coefficient of 5600 $M^{-1} cm^{-1}$ and protein purity was judged by LC-MS.

*Production of AlexaFluor594-tagged α-SYN PFFs.* Following the protocol developed by Pinotsi and colleagues[84] (Fig. 5a), A90C α-SYN was labelled with maleimide-modified Alexa Fluor 594 dye (ThermoFisher Scientific) via the cysteine thiol moiety. The labelled protein was purified from the excess of free dye by dialysis against phosphate-buffered saline (PBS) at 4 °C overnight, divided into aliquots, flash frozen in liquid $N_2$ and stored at −80 °C.

PFFs were formed by incubating 180 µM of WT α-Syn with 20 µM of A90C α-Syn (final concentration of 200 µM monomeric protein) (PBS, pH 7.4) at 45 °C, stirring with a micro-stirrer. At 24 h intervals, the fibril solution was sonicated using a water bath sonicator for 15 min. After 72 h, the fibril solutions were divided into 50 µM aliquots, flash frozen with liquid $N_2$ and stored at −80 °C until required. The efficiency of the labelling process was checked by LC-MS at the Bioanalytical Mass Spectrometry Facility of the UNSW. This was confirmed by treating the solution of fibrils with an enzymatic detergent at 60 °C for 30 min to achieve full disaggregation into labelled monomer and by comparing the resulting fluorescence trace with known concentrations of free dye. Labelling efficiency of the α-SYN monomer was determined to be ~10% (i.e. 10% of the monomers were labelled), consistent with Pinotsi et al.[84].

For the single-molecule experiments, the solutions were diluted to 5 µM (monomer-equivalent) in PBS and sonicated for a further 10 min just before use to obtain a homogeneous distribution of sizes, as described elsewhere[84,85]. Two-colour single-molecule coincidence experiments were carried out as described above and apparent stoichiometry of interactions with fibrils was calculated.

*Cell-free protein expression and co-expression.* Briefly, throughout this work 4 types of experiments were performed representing 4 different modalities of cell-free expression of proteins on LTE: (a) single protein expression levels, either GFP or mCherry tagged; (b) simple co-expression of ~1:1 expression levels of proteins tagged with each of the fluorescent tags; (c) single protein expression levels followed by mixing of the two expression reactions; (d) single protein expression levels followed by mixing with purified PPFs of α-SYN.

a. For single protein expression levels, cell-free expression was carried out by adding DNA to LTE in a ratio of 1:9 and 2:8 for GFP- and mCherry-tagged proteins, respectively. This is because mCherry tags typically display a slower folding time, resulting in overall lower expression levels than GFP-tagged proteins. Proteins were allowed to express for 2.5 h at 27 °C, followed by 0.5 h at 37 °C.

b. Co-expression of GFP and mCherry-tagged proteins were performed in the respective ratios of 20 and 40 nM of DNA template, for a total of 10 μL of LTE. Proteins were allowed to co-express for 2.5 h at 27 °C, followed by 0.5 h at 37 °C.

c. Mixing two separate GFP- and mCherry-tagged proteins after individual expression was performed to understand modes of binding for identified interactors. Individual expression levels were carried out as above, and mixing was done at 1:1 v/v of the expression reactions after 2 h of expression. Mixed samples were allowed to rest at room temperature for 30 m before microscope measurements.

d. Finally, we also assessed the binding of N-GFP LB proteins to PPFs of α-SYN. Stocks of purified α-SYN fibrils were stored, with 100 μM monomer-equivalent concentration of fibrils. Fibrils were then diluted 1:10 in LTE containing the DNA of LB proteins. Expression was allowed to occur for 2.5 h at 27 °C, and 0.5 h at 37 °C, in the presence of fibrils. The reaction was then diluted 1:10 for experiments, to reach a fibril concentration of 1 μM (monomer-equivalent).

All samples acquired as described above were diluted 1:10 in 25 mM HEPES and 50 mM NaCl directly in the sample holder for microscope measurements.

*Sample preparation and microscope set-up.* Samples obtained through cell-free expression, as described above, were immediately loaded into a custom-made 192-well silicone plate with a 70 × 80 mm glass coverslip (ProSciTech, Kirwan, QLD, Australia). Plates were analysed at room temperature on Zeiss Axio Observer microscope (Zeiss, Oberkochen, Germany) with a custom-built data acquisition set-up. Illumination is provided by a 488 nm and a 561 nm laser beams, co-focussed in the sample volume using a ×40 magnification, 1.2 Numerical Aperture water immersion objective (Zeiss, Oberkochen, Germany). This creates a very small observation volume in solution (~1 fL), through which fluorescent proteins diffuse, emitting light in specific wavelengths as their fluorescent tags are excited by the laser beams. Light emitted by the fluorophores is split into GFP and mCherry channels by a 560 nm dichroic mirror (Dichroic 3). The fluorescence of GFP is measured through a 525/50 nm band-pass filter and the fluorescence of mCherry is measured through a long-pass filter. Fluorescence is detected by two photon counting detectors (Micro Photon Devices, Bolzano, Italy). Photons of the two channels are recorded simultaneously in 1 ms time bins and analysed using Lab-VIEW 2018 version 18.0 (National Instruments). For experiments with rarer aggregates, in order to increase the efficiency of event detection, the plate holder of the microscope was adapted to move at a constant set speed during acquisition. This step allowed us to retrieve a high number of events, even under excess of monomer.

*TCCD to study protein–protein interactions.* We used the microscope set-up described above to excite GFP- and Cherry/AF594-tagged proteins in a cell-free expression system. The identification of fluorescent events in TCCD was achieved by counting the number of photons emitted at a set interval time τ. The presence of two events on both channels at the same bin time (we used 10 ms bin time) is read as a coincident event.

To perform all our TCCD experiments, we followed an optimized methodology published by Clarke and colleagues[12] to analyse the co-expression traces and measure co-diffusion between dual-labelled molecules, as described next.

## Association Quotient (Q)

Appropriate thresholds were calculated automatically by plotting the population of Association Quotients (Q) values as a function of the thresholds and finding the maximum Q values. Q is defined as:

$$Q = \frac{(C - E)}{(A + B - (C - E))} \quad (1)$$

where A and B are the events in the two channels, the observed rate of coincident events is C and the estimated rate of events that occur by chance is given by E, which in turn can be defined as:

$$E = AB\tau \quad (2)$$

where τ is the interval time in seconds. We used the approach described in the manuscript mentioned above[12] to calculate for each time trace of co-expressed m-Cherry- (or AF594-fibrils) and GFP-tagged proteins the optimal threshold, as shown in Fig. S3. For each sample, four independent measurements (traces) were acquired and Q values were averaged, to give us an average Q value for a specific interaction. We obtained average Q values for co-expression between N-GFP-tagged LB proteins and C-mCherry α-α-SYN. We used the corresponding α-SYN C-GFP-tagged as a positive control; co-expression of C-mCherry α-SYN with GFP monomer served as a negative control. For the fibril-binding assays, we used the maximum Q value for normalization. Average Q values were used as a measure of interactions between protein pairs and BIs were calculated by normalizing against the maximum and minimum values detected (positive and negative controls). BIs were used to obtain heatmaps of interactions.

**Scanning well measurements.** To increase the number of co-aggregates detected by TCCD, we measured fluorescence in a "scanning well" mode. In this acquisition mode, the detection volume is fixed, but the sample is slowly translated horizontally at a speed of 10 μm/s. This enables the detection of slowly diffusing particles and the scanning of a larger volume of sample, as in ref. [15]. To avoid measuring the same particles multiple times, the well is translated in x-direction for 15 s, then shifted by 200 μm in the y-direction, scanned in x in the reverse direction for 15 s, moved another 200 μm in the same y-direction and so on. The slow translation of the stage does not affect the measurement of binding and stoichiometries.

**Purification of oligomers of WT α-SYN.** WT α-syn oligomers were generated and purified based on the protocols by Kumar et al.[14] and Rösener et al.[86]. Briefly, purified monomeric WT α-SYN was lyophilized and resuspended in 600 μL PBS to a final α-syn concentration of 12 mg/mL. The reaction was placed at 37 °C while shaking at 900 r.p.m. and stopped after 5 h of incubation. The solution was centrifuged at $18,000 \times g$ for 10 min at 4 °C to remove large fibrillar species. The oligomers and monomeric proteins were separated by size-exclusion chromatography using a Superdex 200 Increase 10/300 GL in PBS (GE Healthcare, 28990944). The chromatogram revealed two elution peaks: at 8.5 and 14.5 mL corresponding to oligomers and monomers in agreement to what has been reported[14,86]. As expected, the 15 mL elution peak contains the monomeric α-SYN, while the first elution peak contains oligomers. Negative staining of these fractions reveals spherical particles with a narrow distribution of 12.4 ± 2.5 nm in diameter[87]. The oligomeric fractions were kept separate and flash-frozen in liquid nitrogen. The oligomer fractions used in this study had a monomeric α-SYN concentration of 2 μM, as estimated using reducing SDS-PAGE gel densitometry with purified recombinant α-SYN as standard.

**Binding of interactors to the purified oligomers.** Interactors were expressed for 2 h 30 min in LTE at 27 °C and then diluted in AlphaScreen buffer (2.5 μL of cell-free extracts added to 15 μL of buffer). Three fluorescence traces of 100 s were acquired for each interactor to establish the baseline behaviour in the absence of oligomers. For each trace, the brightness parameter was calculated as $B = \sigma^2/\mu$, where σ is the standard deviation of the fluorescence intensities acquired in the trace, and μ is the average value of fluorescence.

In all, 2.5 μL of purified oligomers were then added to the diluted interactors, incubated for 1 min, then 3 fluorescent traces of 100 s were acquired and B values were calculated as above. Upon binding to oligomers, multiple interactors are brought together in the same complex, creating brighter diffusing particles, therefore, brightness is a sensitive parameter to detect binding to unlabelled particles[15,88]. Monomeric GFP was used as a control, validating that brightness values were unchanged upon mixing with the oligomers. In contrast, large changes were observed for CRYAB and proteins from the GABARAP family, demonstrating that these proteins can bind pre-formed purified oligomers (Fig. 4).

**Single-molecule characterization of cell-free expressed α-SYN mutants using AttoBright.** A30P, G51D and A53T α-Syn tagged with a GFP at the C-terminus were expressed in cell-free as described before. Samples were diluted 1 in 10 (2 μL + 18 μL of AlphaScreen buffer) in a custom PDMS plate adhered to a glass coverslip (thickness #1.5). The samples were analysed for aggregation on our three-dimensional-printed confocal microscope ("AttoBright" set-up[89,90]) equipped with a 450 nm laser and water immersion ×40/1.2 NA objective (Zeiss). Emitted fluorescence from GFP was transmitted through a dichroic mirror (488 nm) and a long-pass filter (500 nm) before focussing on single-photon avalanche diode (MPD Bolzano). Fluorescence spectroscopy traces were recorded for 300 s/trace in 10 ms bins and analysed using a custom python script[90].

**Statistics and reproducibility**
*Replication.* The in vitro experiments described in this manuscript were performed with three or more biological replicates, i.e. separate expression levels with different cell-free extracts. LTE extracts have demonstrated reproducibility in terms of expression levels and binding or co-aggregation behaviours. Multiple batches of cell-free lysates were tested to ensure that the results are reproducible.

*Sample size.* In our in vitro experiments, binding is calculated over hundreds to thousands of individual proteins. The duration of the acquisition and concentrations of the proteins were calibrated to yield sufficient number of binding/co-aggregation events. In the case of co-aggregation with cell-free expressed oligomers, the scanning well method was used to increase the number of events, as described in the text and in the Supplementary Information.

**Reporting summary.** Further information on research design is available in the Nature Research Reporting Summary linked to this article.

## Data availability

The Source data for graphs and charts is available as Supplementary Data 1 and any remaining information can be obtained from the corresponding author upon reasonable request.

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

## Acknowledgements

A.D.G.L. was supported by the EMBL Australia Partnership PhD Program. This work was also supported by grants from the National Health and Medical Research Council of Australia (project grants APP1108859 to Y.G. and E.S., APP1120374 to E.S.). Y.G. was supported by an Australian Research Council Future Fellowship (FT110100478) during this project.

## Author contributions

A.D.G.L., Y.G. and E.S. designed the research. A.D.G.L. performed cell-free protein expression and co-expression, purification and labelling of α-synuclein, production of pre-formed fibrils of α-synuclein and two-colour coincidence experiments. D.J.B.H. performed gateway cloning of the library of Lewy body protein constructs. P.R.-S. and A.B. performed AlphaScreen experiments. A.B. helped with purification of α-synuclein. A.C. helped with construct preparation and AlphaScreen assays. D.J.B.H. produced all cell-free lysates used in this study. D.L. performed purification of α-synuclein oligomers. Y.G. and E.S. performed binding assays to purified oligomers and directed the research. A.D.G.L., Y.G. and E.S. wrote the manuscript and all authors contributed to the revisions of the manuscript.

## Competing interests

The authors declare no competing interests.
