## [Transparent Peer Review File · Communications Biology]

Reviewers' comments:

Reviewer #1 (Remarks to the Author):

Gambin and co-workers describe an impressive tour de force analysis of how different members of the Lewy Body proteome interact with various aggregated species of alpha-syn. I was generally very impressed by the thoroughness, clarity and very systematic approach taken – the authors even went so far as to compare with the interactome of another aggregation-prone protein PINK1. This work will be of great interest to many readers and I just have a few suggestions for improvement.

1. The weakest point is the definition of the oligomeric state. The screening for oligomer binding is ingenious (comparison of different mutants with different aggregation propensities) but the “oligomer” status is poorly defined. The authors simply classify oligomers as a fluorescent burst well above background. This just describes a broad range of soluble alpha-syn complexes without reference to specific species. How broad is this distribution of bursts (i.e. oligomer size)? It is a strength on the one hand that they allow alpha-syn to aggregate spontaneously but given the variety of different species (some on-pathway, others off-pathway) it is difficult to know if there are subpopulations of “attractors” for different types. It would be more convincing if separately prepared oligomers of alpha-syn could be prepared and added to the mixture as well-defined species. I appreciate that the authors have actually done an experiment along these lines (“Coaggregation versus Binding to pre-formed aggregates”) but I could not find any description of how these aggregates were formed under Methods nor see any description of their properties (size etc).

2. Several groups have published work on the interactomes of aSN fibrils and oligomers (see e.g. work by the Melki, Otzen and Jensen groups). It would be a strong benefit for the reader to compare the present results with those previously published. Given the authors’ admirably systematic approach to the handling of large bodies of data, I am sure this can be presented in a clear and enlightened manner.

Minor issues:

1. P. 5: “This method is described by Clarke and colleagues¹¹ and was throughout this study (Figure S3).” Presumably “used” is missing?

Reviewer #2 (Remarks to the Author):

To better understand the specific cascade of protein interactions that lead to the pathological aggregation of a-synuclein (wild-type and mutant forms), the authors present a novel method to measure interactions of monomeric, oligomeric, and fibrillar forms of a-syn with 65 proteins that were previously shown to be components of Lewy bodies. They expressed individual GFP-tagged partner proteins, and mCherry-tagged a-syn, in a cell free system, and then measured protein interactions directly in the cell lysates, bypassing the need to purify the proteins individually. They employed a bead-transfer assay to look at interactions of partner proteins with monomeric a-syn, and an optimized version of confocal fluorescence spectroscopy to measure interactions with oligomeric and fibrillar a-syn. They present evidence that some partner proteins interact preferentially with monomers and small oligomers, while others seem to co-aggregate with a-syn, showing minimal binding to pre-formed fibrils. More specifically, they find that members of the ATG8 protein family bind to pre-fibrillar aggregates of a-syn, and they discuss a potential mechanism whereby these interactions could disrupt autophagic clearance of misfolded protein, potentially leading to disease. They also demonstrate a degree of specificity of the interactions of partner proteins with a-syn fibrils, as different sets of proteins show preferential binding to a-syn fibrils versus fibrils formed from another protein protein to self-aggregation, PINK1.

This is a novel approach to understanding the sequence of protein-binding events that lead to a-syn aggregation. The data are convincing, and the use of PINK1 fibrils as a control for specificity lends extra confidence that the results are meaningful.

A few minor suggestions:

page 10, have not defined PPI (presumably protein-protein interactions)

page 12, have not defined ALP in the phrase "autophagic flux reduced ALP impairment"

Figure 3a illustrates the derivation of G, R, and C, but E is only defined in the legend

Figure 7 says that "distance from center represents change in binding index"; might be clearer to say "magnitude of binding index"

Reviewer #3 (Remarks to the Author):

In the manuscript entitled "Selectivity of Protein Interactions along the Aggregation Pathway of α -Synuclein", Leitão et al. characterized the interactions of alpha-synuclein (α -SYN) in vitro using established kinetoplastid cell-free lysate and bacterial protein expression systems. Specifically, they claim to have implemented novel single-molecule methods to study co-aggregation and binding events between α -SYN and other proteins present in a pathological hallmark of Parkinson's disease (PD), inclusions named as Lewy Bodies (LB). They further claim to have painted a novel picture of aggregation cascades and protein-protein interactions that lays the groundwork for future studies on α -SYN aggregation. While this reviewer appreciates the interaction landscape depicted in this study, there are significant problems (as outlined below) associated with this work that makes it unattractive to a wider readership in its current form.

1. The work is extremely empirical in nature and this reviewer fails to recognize the novelty claimed by the authors. The methods used in this work are all previously published and many of the experiment-specific modifications are also adopted from previously published work. Further, the candidate hits from the interaction experiments are also largely well established. So, a novelty in the aggregation cascade as the authors have claimed is also lacking. I suggest the authors should thoroughly revise the manuscript text to resolve these claims. Communications Biology is not strictly focused on the impact or significance of findings, so there is nothing wrong with reporting empirical findings. It would significantly improve the appeal of the work.

2. It is important to discuss the serious limitations of the work. To start with, the LTE screen used 65 ORFs that were all N-terminal GFP tagged. I understand it is necessary for this particular screen to maintain homogeneity but it is necessary to address this as a significant limitation of the study. The folding pattern of 65 proteins would be largely different from each other and the N-terminal tag for all would have a differential effect on the folding dynamics.

3. The results of proximity-based screens are affected by false positives (especially because alpha screen signals are largely concentration-dependent). It is necessary to validate the final 3 hits of this study by an alternative approach, such as FRET.

4. The co-translational expression and co-folding form the basis of a majority of the experiments in this study and it is important to discuss this in comparison to protein expression and folding in mammalian cells. The concepts of "co-translation" and "co-folding" are wildly different inside mammalian cells.

5. For the Two-color coincidence detection setup, it is necessary to explain further how the plate holder of the microscope was adapted to move at a constant speed during acquisition, how this constant movement of the stage affects data acquisition, what controls were performed to show that

the process does not introduce data acquisition artifacts.

6. The discussions on "Previous studies have shown that membrane-bound-GRK5 not only colocalizes with α -SYN in Lewy Bodies.....", where the authors describe phosphorylation and interactions with membrane-bound forms of GRK5 are largely out of context in this work. The references cited have mostly studied such phenotypes in cellular or in vivo context, much different from the experimental setup of this work.

7. The discussions on autophagy-related hits should be made contextual. The protein interactions identified by in vitro assays may well reflect the possibility of similar intracellular interactions, however, it does not reflect the steps of a hierarchical cellular process that is equally dependent on the sub-cellular microenvironment.

REVIEWS COMMUNICATIONS BIOLOGY

Reviewer #1 (Remarks to the Author):

Gambin and co-workers describe an impressive tour de force analysis of how different members of the Lewy Body proteome interact with various aggregated species of alpha-syn. I was generally very impressed by the thoroughness, clarity and very systematic approach taken – the authors even went so far as to compare with the interactome of another aggregation-prone protein PINK1. This work will be of great interest to many readers and I just have a few suggestions for improvement.

We thank the reviewer for this positive assessment and these comments.

1. The weakest point is the definition of the oligomeric state. The screening for oligomer binding is ingenious (comparison of different mutants with different aggregation propensities) but the “oligomer” status is poorly defined. The authors simply classify oligomers as a fluorescent burst well above background. This just describes a broad range of soluble alpha-syn complexes without reference to specific species. How broad is this distribution of bursts (i.e. oligomer size)?

It is true that “oligomeric” is an ill-defined term, especially in the field of synuclein. We previously reported (Sierecki et al., Sci. Rep 2015) that the different mutants formed different sizes of aggregates upon expression, as defined by the intensity of the fluorescent bursts. As the synuclein mutants are GFP-labelled, the intensity of the fluorescent burst is proportional to the number of proteins in the assembly. A53T aggregates were found to be different than G51D and A30P complexes.

To further define these objects, we added a new experiment to quantify both the intensity and residence time of the fluorescent events. This analysis (published in Bhumkar et al., Angw. Chem. 2021) reveals differences between mutants in more details and provides an overview of the distribution of the bursts.

We added a new supplementary figure S4. and included the following statement in the manuscript:

“Analysis of the maximal intensity and residence time of each fluorescent burst shows that the objects formed by the different mutants are relatively homogeneous and confirms that A30P aggregates are less fluorescent (and therefore contain less proteins per aggregates) than the ones formed by G51D and A53T (Figure S4).” (p 6).

Figure S4: fingerprinting of the three Synuclein mutants expressed in cell-free.

A30P, G51D and A53T α -SYN mutants were expressed in our cell-free system with a C-terminal GFP tag and the size of the aggregates was characterized using a dedicated 3D printed microscope⁹⁰. On this instrument, each Synuclein aggregate can be analysed for its residence time in the focal volume, using a recently developed algorithm⁹¹. **a-c** Raw fluorescence trace obtained for the three mutants on the AttoBright setup; the binning time of the fluorescence intensity is 10ms. **d-f** Analysis of diffusion time (residence time in the focal volume) and maximal intensity of the peak for the three mutants. The difference colours correspond to replicates. **g-i** Violin plots of the maximal Intensity, diffusion times and area under the curve for the three mutants.

And in Material and Methods

“Single molecule characterisation of cell-free expressed α -SYN mutants using AttoBright:

A30P, G51D and A53T α -Syn tagged with a GFP at the C-terminus were expressed in cell-free as described before. Samples were diluted 1 in 10 (2 μ L + 18 μ L of AlphaScreen buffer) in a custom PDMS plate adhered to a glass coverslip (thickness #1.5). The samples were analysed for aggregation on our 3D-printed confocal microscope (“AttoBright” setup^{90,91}) equipped with a 450 nm laser and water immersion 40x/1.2 NA objective (Zeiss). Emitted fluorescence from GFP was transmitted through a dichroic mirror (488 nm) and a long-pass filter (500 nm) before focusing on single photon avalanche diode (MPD Bolzano).

Fluorescence spectroscopy traces were recorded for 300 sec/trace in 10 ms bins and analysed using a custom python script⁹¹.”

It is a strength on the one hand that they allow alpha-syn to aggregate spontaneously but given the variety of different species (some on-pathway, others off-pathway) it is difficult to know if there are subpopulations of “attractors” for different types. It would be more convincing if separately prepared oligomers of alpha-syn could be prepared and added to the mixture as well-defined species. I appreciate that the authors have actually done an experiment along these lines (“Coaggregation versus Binding to pre-formed aggregates”) but I could not find any description of how these aggregates were formed under Methods nor see any description of their properties (size etc).

To validate the binding to “well-defined” oligomers, we purified oligomers of recombinant synuclein according to Kumar et al. (J. Neurochem. 2020) and tested the binding of 8 proteins including 5 “co-aggregators” of our cell-free expressed synuclein mutant oligomers. The data show that GABARAP, GABARAPL1 and GABARAPL2 interact with recombinant, wild type synuclein oligomers.

We included new panels d and e of Figure 4, a new supplementary Figure S7 and amended the manuscript:

“To further validate that a subset of proteins were bona fide oligomer interactors, we generated and purified oligomers from recombinant wild-type α -SYN, according to Kumar et al.¹⁵ Oligomers were separated by size-exclusion chromatography and characterized by electron microscopy. To test binding to these unlabelled objects (**Figure 4d**), we observed changes in the aggregation behaviour of the interactors and quantified the brightness of the GFP-labelled interactor before and after addition of unlabelled oligomers¹⁶. Binding to oligomers was detected by the apparition of fluorescent bursts in the GFP channel shortly after mixing (**Figure S7**). As shown in **Figure 4e**, we observed an increase of brightness for GABARAP, GABARAPL1 and GABARAPL2 upon mixing with oligomers. Addition of oligomers had no effect on the controls GFP, CHAT and PARK7 which were not identified as oligomer interactors.” (p 7)

The Material and methods section was also modified accordingly.

“Purification of oligomers of WT α -SYN

WT α -syn oligomers were generated and purified based on the protocols by Kumar et al.¹⁵ and Rösener et al.⁸⁸. Briefly, purified monomeric WT α -SYN was lyophilised and resuspended in 600 μ L PBS to a final α -syn concentration of 12 mg/mL. The reaction was placed at 37 °C while shaking at 900 rpm and stopped after 5h of incubation. The solution was centrifuged at 18,000g for 10min at 4°C to remove large fibrillar species. The oligomers and monomeric proteins were separated by size-exclusion chromatography using a Superdex 200 Increase 10/300 GL in PBS (GE Healthcare, 28990944). The chromatogram revealed two elution peaks: at 8.5 mL and 14.5 mL corresponding to oligomers and monomers in agreement to what has been reported^{15, 88}. As expected, the 15mL elution peak contains the monomeric α -SYN, while the first elution peak contains oligomers. Negative staining of these fractions reveals spherical particles with a narrow distribution of 12.4 ± 2.5 nm in diameter (data not shown). The oligomeric fractions were kept separate and flash-frozen in liquid nitrogen. The oligomer fractions used in this study had a monomeric α -SYN concentration of 2 μ M, as estimated using reducing SDS-PAGE gel densitometry with purified recombinant α -SYN as standard.

Binding of interactors to the purified oligomers:

Interactors were expressed for 2h30 in LTE at 27°C then diluted in AlphaScreen buffer (2.5 µL of cell-free extracts added to 15 µL of buffer). 3 fluorescence traces of 100s were acquired for each interactor, to establish the baseline behaviour in the absence of oligomers. For each trace, the Brightness parameter was calculated as $B = \sigma^2 / \mu$, where σ is the Standard Deviation of the fluorescence intensities acquired in the trace, and μ is the average value of fluorescence.

2.5 µL of purified oligomers were then added to the diluted interactors, incubated for 1min, then 3 fluorescent traces of 100 s were acquired and B values were calculated as above. Upon binding to oligomers, multiple interactors are brought together in the same complex, creating brighter diffusing particles, therefore, Brightness is a sensitive parameter to detect binding to unlabeled particles ^{16, 89}. Monomeric GFP was used as a control, validating that Brightness values were unchanged upon mixing with the oligomers. In contrast, large changes were observed for CRYAB and proteins from the GABARAP family, demonstrating that these proteins can bind pre-formed purified oligomers (**Figure 4**)."

New Figure 4 d Cartoon depicting mixing interactors with purified oligomers of α -SYN. Red circles represent unlabelled WT α -SYN oligomers and green dots represent GFP-tagged Lewy Body components. **e** Bar plots of Brightness values of the GFP-tagged interactors in the presence (orange) or absence (grey) of purified oligomers.

Figure S7. Kinetics of binding of GABARAPL2 to purified oligomers

(top): Raw trace of fluorescence measured for monomeric GFP expressed in cell-free, diluted in AlphaScreen buffer. The protein was measured diluted in buffer, then recording was blocked during addition and mixing of the purified oligomers. The last part of the measurement corresponds to GFP in the presence of oligomers. The GFP protein remains monomeric and B values remain unchanged (see **Figure 4**).

(bottom): raw fluorescence trace obtained for GABARAPL2 expressed in cell-free with a GFP tag at the N-terminus, before and after addition of purified oligomers. Brighter peaks appear only after addition of the oligomers, showing that multiple GABARAPL2 are recruited on the same particle and co-diffuse. This binding causes an increase in the Brightness values, as shown in **Figure 4**. Note that the Brightness value is concentration-independent in the range of intensities used here (between 400 photons/ms and 5000 photons/ms)^{92, 93}.

2. Several groups have published work on the interactomes of aSN fibrils and oligomers (see e.g. work by the Melki, Otzen and Jensen groups). It would be a strong benefit for the reader to compare the present results with those previously published. Given the authors' admirably systematic approach to the handling of large bodies of data, I am sure this can be presented in a clear and enlightened manner.

The interactors identified in these studies did not overlap in general with the proteins tested here. Nevertheless, we included these findings in the discussion.

The discussion was modified:

"We believe that the main contribution of this study is an unbiased deconvolution of monomeric, oligomeric and fibrillar interactors of α -SYN. Our study, as a purely *in vitro* approach, does not tell a story of the interactome of synuclein inside the cell, nor does it intend to. It nonetheless suggests that recruitment of protein partners is hierarchical, occurring at specific steps along the aggregation of α -SYN from monomer

to amyloid fibrils. This hypothesis, though still controversial, had been proposed before by Betzer and colleagues²⁴, whose work described a top-down mass spectrometry approach using monomeric and oligomeric α -SYN as baits in lysed fractions of porcine brains. Despite large differences in datasets and approaches, we validated interactions between α -SYN oligomers and $\alpha\beta$ -crystallin (CRYAB) as well as between A53T oligomers and 60-kDa heat shock protein HSPD1. Our work expands this dataset by showing that pre-formed amyloid fibrils of α -SYN also act as recruiters of proteins (passive or active as this recruitment may be). Our results suggest that α -SYN oligomers and fibrils are the main contributors to the complexity of protein interactions in Lewy Bodies, whereas monomeric α -SYN is not as promiscuous an IDP as previously thought²⁵. The study by Betzer and colleagues²⁴ corroborates this idea, as only ten proteins were pulled-down with the α -SYN monomers, in contrast with 76 interactors of oligomers.” (p10)

“ α -SYN interactome also depends on the cellular context²⁹. A recent study using APEX labelling of α -SYN in neurons coupled with mass spectrometry has identified 225 proteins, including synaptic proteins, proteins involved in endocytic vesicle trafficking, the retromer complex, phosphatases and mRNA binding proteins³⁰. A follow-up study showed that endocytic trafficking proteins overlapped with genetic modifiers of α -SYN toxicity, suggesting that perturbation of this pathway is directly related to the spatial localization of α -SYN within the cell³¹. While here we did not test proteins involved in endocytosis specifically, the high interaction scores between α -SYN aggregates (oligomers and fibrils) with proteins from the autophagy-lysosomal pathway, suggests that aggregation might act a disrupter of normal vesicle formation, coalescence, or trafficking. Interestingly, the same study found high enrichment of both MAP1LC3A (homolog of MAP1LC3B) and MAPT³⁰, which we show here to co-diffuse with pre-formed synuclein oligomers and fibrils. As mentioned, the primary goal of this study was to show that self-assembly affects protein partner recognition. This in-vitro study has limitations (discussed below) as it fails to recapitulate the complexity of what is happening in a cellular context, from the role of post-translational modifications to the importance of cellular localization and micro-environment. Nevertheless, we believe that it also sheds a light on previous observations in relevant cellular context, due to its inherent simplicity. Examples of such insights are presented below. The importance of these kind of studies has been recently demonstrated by Melki and colleagues³² who engineered polypeptides to coat the surface of α -SYN fibrils in such a way that their ability to bind to the plasma membrane components and/or their take-up by cell are diminished.” (p11)

Minor issues:

1. P. 5: “This method is described by Clarke and colleagues and was throughout this study (Figure S3).” Presumably “used” is missing?

We corrected this typo in the manuscript.

Reviewer #2 (Remarks to the Author):

To better understand the specific cascade of protein interactions that lead to the pathological aggregation of α -synuclein (wild-type and mutant forms), the authors present a novel method to measure interactions of monomeric, oligomeric, and fibrillar forms of α -syn with 65 proteins that were previously shown to be components of Lewy bodies. They expressed individual GFP-tagged partner proteins, and mCherry-tagged α -syn, in a cell free system, and then measured protein interactions directly in the cell lysates, bypassing the need to purify the proteins individually. They employed a bead-transfer assay to look at interactions of partner proteins with monomeric α -syn, and an optimized version of confocal fluorescence spectroscopy to measure interactions with oligomeric and fibrillar α -syn. They present evidence that some partner proteins interact preferentially with monomers and small oligomers, while others seem to co-aggregate with α -syn, showing minimal binding to pre-formed fibrils. More specifically, they find that members of the ATG8 protein family bind to pre-fibrillar aggregates of α -syn, and they discuss a potential mechanism whereby these interactions could disrupt autophagic clearance of misfolded protein, potentially leading to disease. They also demonstrate a degree of specificity of the interactions of partner proteins with α -syn fibrils, as different sets of proteins show preferential binding to α -syn fibrils versus fibrils formed from another protein protein to self-aggregation, PINK1. This is a novel approach to understanding the sequence of protein-binding events that lead to α -syn aggregation. The data are convincing, and the use of PINK1 fibrils as a control for specificity lends extra confidence that the results are meaningful.

We thank the reviewer for this assessment.

A few minor suggestions:

-page 10, have not defined PPI (presumably protein-protein interactions)

We changed PPI to protein-protein interactions in this manuscript.

-page 12, have not defined ALP in the phrase "autophagic flux reduced ALP impairment"

We incorporated the definition of ALP (autophagy-lysosomal-pathway).

-Figure 3a illustrates the derivation of G, R, and C, but E is only defined in the legend

E represents the chance of coincidence. We added the mathematical definition of the chance of coincidence E in Supplementary data. E is calculated as the number of green events above threshold times the number of red events above threshold, multiplied the interval time in seconds.

The definition of E was added to Figure 3a (iii) and included in the supplementary material.

-Figure 7 says that "distance from center represents change in binding index"; might be clearer to say "magnitude of binding index"

We modified the caption of Figure 7. "Proteins are grouped by biological process and distance to the center represents magnitude of Binding index."

Reviewer #3 (Remarks to the Author):

In the manuscript entitled “Selectivity of Protein Interactions along the Aggregation Pathway of α -Synuclein”, Leitão et al. characterized the interactions of alpha-synuclein (α -SYN) *in vitro* using established kinetoplastid cell-free lysate and bacterial protein expression systems. Specifically, they claim to have implemented novel single-molecule methods to study co-aggregation and binding events between α -SYN and other proteins present in a pathological hallmark of Parkinson’s disease (PD), inclusions named as Lewy Bodies (LB). They further claim to have painted a novel picture of aggregation cascades and protein-protein interactions that lays the groundwork for future studies on α -SYN aggregation. While this reviewer appreciates the interaction landscape depicted in this study, there are significant problems (as outlined below) associated with this work that makes it unattractive to a wider readership in its current form.

We have improved the manuscript based on this feedback, as outlined.

1. The work is extremely empirical in nature and this reviewer fails to recognize the novelty claimed by the authors. The methods used in this work are all previously published and many of the experiment-specific modifications are also adopted from previously published work. Further, the candidate hits from the interaction experiments are also largely well established. So, a novelty in the aggregation cascade as the authors have claimed is also lacking. I suggest the authors should thoroughly revise the manuscript text to resolve these claims. Communications Biology is not strictly focused on the impact or significance of findings, so there is nothing wrong with reporting empirical findings. It would significantly improve the appeal of the work.

We recognize the empirical nature associated with a large screen of protein-protein interactions such is the one described in this work. Our approach was indeed an agnostic screen of interactions, focussing on selectivity rather discovering new synuclein interactors. In this study, we validated our central hypothesis that the stage of self-assembly of alpha-synuclein dictates differential binding events with other proteins that have previously been shown to co-immunoprecipitate with alpha-synuclein in Lewy Bodies lesions. We believe that the selectivity we demonstrate is a significant contribution to the field. We agree that the physiological relevance of the specific interactions identified here will require further validation. We have listed all the limitations of this study in the discussion.

We modified the discussion to highlight the added value of our approach and also discuss the limits of our findings. We added:

“ We believe that the main contribution of this study is an unbiased deconvolution of monomeric, oligomeric and fibrillar interactors of α -SYN. Our study, as a purely *in vitro* approach, does not tell a story of the interactome of synuclein inside the cell, nor does it intend to. It nonetheless suggests that recruitment of protein partners is hierarchical, occurring at specific steps along the aggregation of α -SYN from monomer to amyloid fibrils. This hypothesis, though still controversial, had been proposed before by Betzer and colleagues²⁵, whose work described a top-down mass spectrometry approach using monomeric and oligomeric α -SYN as baits in lysed fractions of porcine brains. Despite large differences in datasets and

approaches, we validated interactions between α -SYN oligomers and $\alpha\beta$ -crystallin (CRYAB) as well as between A53T oligomers and 60-kDa heat shock protein HSPD1. Our work expands this dataset by showing that pre-formed amyloid fibrils of α -SYN also act as recruiters of proteins (passive or active as this recruitment may be). Our results suggest that α -SYN oligomers and fibrils are the main contributors to the complexity of protein interactions in Lewy Bodies, whereas monomeric α -SYN is not as promiscuous an IDP as previously thought²⁶. The study by Betzer and colleagues²⁵ corroborates this idea, as only ten proteins were pulled-down with the α -SYN monomers, in contrast with 76 interactors of oligomers.” (p10)

“As mentioned, the primary goal of this study was to show that self-assembly affects protein partner recognition. This in-vitro study has limitations (discussed below) as it fails to recapitulate the complexity of what is happening in a cellular context, from the role of post-translational modifications to the importance of cellular localization and micro-environment. Nevertheless, we believe that it also sheds a light on previous observations in relevant cellular context, due to its inherent simplicity. Examples of such insights are presented below. The importance of these kind of studies has been recently demonstrated by Melki and colleagues³³ who engineered polypeptides to coat the surface of α -SYN fibrils in such a way that their ability to bind to the plasma membrane components and/or their take-up by cell are diminished.” (p11)

We also modified some claims of novelty: “Overall, this work ~~paints a novel picture of~~ provides new insights into the protein-protein interactions landscape along the aggregation cascades ~~and~~ and lays the groundwork for future studies on how to modulate α -SYN aggregation at different steps.” (p15)

“Here we provide evidence of direct interactions between members of the GABARAP/LC3 subfamily of autophagosomal proteins and oligomeric/fibrillar α -SYN. Impairment of the formation of autophagosomes by α -SYN aggregates has been shown by several studies, resulting in traffic jams of proteins, organelles and lipid membranes⁶⁶⁻⁶⁸. Previous proteomics studies have provided contradictory results as to whether LC3 proteins actually interact with α -SYN⁶⁹ and our data suggest that such interactions can only occur when aggregation is under way. Stykel et al.⁷⁰ showed direct interaction and sequestration of MAP1LC3B into microaggregates by A53T and E46K mutants of α -SYN, in agreement with what we observed”. (p13)

2. It is important to discuss the serious limitations of the work. To start with, the LTE screen used 65 ORFs that were all N-terminal GFP tagged. I understand it is necessary for this particular screen to maintain homogeneity but it is necessary to address this as a significant limitation of the study. The folding pattern of 65 proteins would be largely different from each other and the N-terminal tag for all would have a differential effect on the folding dynamics.

We thank the reviewer for suggesting that we address the limitations associated with a general screen of PPIs such as the one described in this work.

The paragraph below is now included in the discussion.

“Limitations of the study

In our study we used the pathological mutants and the pre-formed fibrils of α -SYN as the representative species of intermediate (oligomeric) and endpoint species of the aggregation cascade, respectively. In respect with the oligomers originated by A30P, G51D and A53T α SYN upon cell-free expression, it is still largely unclear whether these oligomeric structures represent an on- or off-pathway stage in the aggregation process. Therefore, although the study of pathological mutants provided us with an expedite way to acquire the interactome of different aggregated conformers of α -SYN, the feasibility of associating

our results with the aggregation pathway of the WT protein is, in large part, speculative. However, to study the interactome of the pathological mutants also brings about possibilities to access pathological folds of synuclein and to understand which protein partners in the LBs could be their targets.

Regarding the *in vitro*-generated pre-formed fibrils of α -SYN, despite allowing us to obtain an interactome of α -SYN fibrils with LB proteins, the full biological implications of this interactome are unclear. For example, recent studies describe important differences between PFF's generated *in vitro* from the ones collected from patient brain samples⁷⁶. Furthermore, various polymorphs of α -SYN aggregates occur *in cellulo* and *in vivo*, with different conformations, membrane binding and seeding abilities⁷⁷, and sometimes even leading to different pathologies where Lewy Bodies co-occur⁷⁸. For example, a recent study used liquid chromatography and tandem mass spectrometry to find at least 20 different forms of α -SYN within patient-derived Lewy Bodies⁷⁹. This included acetylated, phosphorylated and truncated α -SYN, although their prevalence in disease is still to be determined.

Another limiting aspect of our study is the fact that it is inherently a simplification of protein-protein interaction networks in the highly heterogeneous environment that is the Lewy Body. For pragmatic reasons we tested only pairwise interactions between Cherry- or AF594-tagged α -SYN conformers and the library of N-GFP tagged LB proteins. This approach does not come without its limitations, as many protein-protein interactions are mediated by cofactors (for example molecular chaperone HSP70 which interacts with alpha-synuclein only when it is released as a substrate from HSP40s) or necessitate the presence of PTMs such as ubiquitination and phosphorylation^{69, 80}.

Finally, the use of GFP and mCherry fluorescent tags attached to one end of the protein is another variable of relative uncertainty, concerning their effect on the interactions that were screened. The relative large size of the tags could theoretically limit the availability of the binding site of a given protein and introduce false negatives into our interactome of α -SYN. Importantly, the presence of a tag should not affect selectivity between the different species of α -SYN. Furthermore, previous studies have demonstrated that tagging synucleins with GFP does not alter their aggregation behaviour⁸¹, despite the small size of the protein. It is possible, however that amongst all the LB proteins tested, some would see their binding abrogated.

Despite this, we believe that the findings of this simplistic approach to deconvolute the interactome of α -SYN in the Lewy Bodies provide important clues into the mechanism of inclusion formation, one pairwise interaction at a time.” (p15)

3. The results of proximity-based screens are affected by false positives (especially because alpha screen signals are largely concentration-dependent). It is necessary to validate the final 3 hits of this study by an alternative approach, such as FRET.

We appreciate the point made about the importance of concentration dependence associated with the nano-bead based proximity assay used in this study. For this reason, we always perform serial dilutions of the co-expressed proteins, in order to find the concentrations that allow for optimal coating of the acceptor and donor beads by the mCherry-synuclein and the GFP-tagged Lewy Body proteins. It is true that false positives are always an issue with protein-protein interaction assays and AlphaScreen is no different. However, our hit rate is (surprisingly) low. Further validation of the hits, by alternative methods, will be required in follow-up studies where individual interactions are examined but we believe this is beyond the scope of this study.

We included a more detailed text in the manuscript.

“AlphaScreen signal is dependent on the concentration of the protein: at optimal protein-to-bead ratios (when the beads are fully coated), a maximum signal is detected as shown by a typical ‘hook effect’ in the

quantified luminescence emitted (see **Figure 2-b** and **Methods**). Therefore, we performed 4 serial dilutions of the samples containing the co-expressed proteins to find the optimal proteins-to-beads ratio. Typically, optimal coating of the beads is achieved when we dilute the LTE by 2-3 orders of magnitude. Because of the high range of concentrations that these dilutions encompass, and since the assay was performed in quadruplicate, false positives should be minimal. Furthermore, at these lower concentrations, dissociation rates are high and even aggregation-prone proteins will be largely monomeric. Therefore, the presence of few aggregates should not overwhelm the system and interactions between aggregates will have the same influence as interactions between two monomers (...). We note that the hit rate of the AlphaScreen experiment is much lower than typically observed in our previous protein-protein interactions screens¹⁰⁻¹²." (p5)

4. The co-translational expression and co-folding form the basis of a majority of the experiments in this study and it is important to discuss this in comparison to protein expression and folding in mammalian cells. The concepts of "co-translation" and "co-folding" are wildly different inside mammalian cells.

We thank the reviewer for pointing this out. Indeed, our use of these terms is incorrect. We removed these words from the text and rephrased when necessary.

We modified the manuscript:

"Coexpression of the proteins increases the probability to detect an interaction. It also allows to detect the formation of heterodimers, when proteins have a tendency to homodimerize." (p4)

5. For the Two-color coincidence detection setup, it is necessary to explain further how the plate holder of the microscope was adapted to move at a constant speed during acquisition, how this constant movement of the stage affects data acquisition, what controls were performed to show that the process does not introduce data acquisition artifacts.

We added details in the Material and Methods section and referenced our recent publication that used the scanning well detection.

"Scanning Well measurements:

To increase the number of co-aggregates detected by TCCD, we measured fluorescence in a "scanning well" mode. In this acquisition mode, the detection volume is fixed, but the sample is slowly translated horizontally at a speed of 10 $\mu\text{m/s}$. This enables the detection of slowly diffusing particles and the scanning of a larger volume of sample, as in¹⁶. To avoid measuring the same particles multiple times, the well is translated in x-direction for 15s, then shifted by 200 μm in the y-direction, scanned in x in the reverse direction for 15s, moved another 200 μm in the same y-direction, and so on. The slow translation of the stage does not affect the measurement of binding and stoichiometries." (p 20)

6. The discussions on "Previous studies have shown that membrane-bound-GRK5 not only colocalizes with α -SYN in Lewy Bodies.....", where the authors describe phosphorylation and interactions with membrane-bound forms of GRK5 are largely out of context in this work. The references cited have mostly studied such phenotypes in cellular

or *in vivo* context, much different from the experimental setup of this work.

Indeed, the references cited relate to cellular or *in vivo* context and depict more complex situations than the *in vitro* assay. We discussed these studies to show that interaction between GRK5 and α -SYN was not unexpected and has a physiological or pathological relevance. Note that this is the case for most of the publications we discuss, as we tried to confront our *in vitro* data with observations in relevant context to establish the significance of our findings.

We explicitly introduced the aim of the discussion and discussed the limits of the comparison in the manuscript, as follows:

“The identification of GRK5 as an interactor of α -SYN is not unexpected based on various observations in cells. Previous studies have shown that membrane-bound-GRK5 not only colocalizes with α -SYN in Lewy Bodies³⁶, but also phosphorylates Ser-129 of α -SYN at the plasma membrane. The membrane localization undoubtedly plays a critical role in the interaction between GRK5 and α -SYN but our *in vitro* data indicate that there is an affinity between the two proteins that contribute to their interaction, beyond co-localization”.

(p15)

And in general “As mentioned, the primary goal of this study was to show that self-assembly affects protein partner recognition. This *in-vitro* study has limitations (discussed below) as it fails to recapitulate the complexity of what is happening in a cellular context, from the role of post-translational modifications to the importance of cellular localization and micro-environment. Nevertheless, we believe that it also sheds a light on previous observations in relevant cellular context, due to its inherent simplicity. Examples of such insights are presented below.” (p11)

7. The discussions on autophagy-related hits should be made contextual. The protein interactions identified by *in vitro* assays may well reflect the possibility of similar intracellular interactions, however, it does not reflect the steps of a hierarchical cellular process that is equally dependent on the sub-cellular microenvironment.

Similarly to before, we tried to clearly indicate that our findings can contribute to the understanding of the processes that occur in cells but do not recapitulate the complexity of what is observed.

We included a general paragraph (p.11, see above) and discussed the importance of post-translational modifications, cellular context and amyloid strains on the interactome in the “limitations of the study” section (p14, as before).

REVIEWERS' COMMENTS:

Reviewer #1 (Remarks to the Author):

The authors have done an excellent job in revising their manuscript in response to the issues I raised. I have no further points that need to be addressed.

Reviewer #3 (Remarks to the Author):

The authors have addressed in detail all of the concerns raised in previous version of the manuscript. I find this manuscript acceptable for publication in Communications Biology.